# Hippocampal-neocortical interactions sharpen over time for predictive actions

Nicholas C. Hindy [1], Emily W. Avery[2] & Nicholas B. Turk-Browne[2]

When an action is familiar, we are able to anticipate how it will change the state of the world. These expectations can result from retrieval of action-outcome associations in the hippocampus and the reinstatement of anticipated outcomes in visual cortex. How does this role for the hippocampus in action-based prediction change over time? We use high-resolution fMRI and a dual-training behavioral paradigm to examine how the hippocampus interacts with visual cortex during predictive and nonpredictive actions learned either three days earlier or immediately before the scan. Just-learned associations led to comparable background connectivity between the hippocampus and V1/V2, regardless of whether actions predicted outcomes. However, three-day-old associations led to stronger background connectivity and greater differentiation between neural patterns for predictive vs. nonpredictive actions. Hippocampal prediction may initially reflect indiscriminate binding of co-occurring events, with action information pruning weaker associations and leading to more selective and accurate predictions over time.

---

[1] Psychological and Brain Sciences, University of Louisville, Louisville, KY 40292, USA. [2] Psychology, Yale University, New Haven, CT 08544, USA. Correspondence and requests for materials should be addressed to N.C.H. (email: nicholas.hindy@louisville.edu)

As you open the door to a familiar room, you are able to anticipate specific objects that will come into view. A neural source of such predictions may be pattern completion in the hippocampus[1–3]. Repeated experience and interaction allows associative learning mechanisms in the hippocampus to bind recurring patterns of objects and actions over space and time[4,5]. Once these links are formed, making an action in response to a familiar cue may prompt the hippocampus to retrieve a conjunctive representation of past events. These representations could contain information about the cue and action, but additionally the yet-to-occur sensory consequences of the action. These retrieved consequences could in turn get reinstated via feedback to sensory systems—a form of memory-based predictive coding of action outcomes.

Decoding of stimulus-related information during action-based prediction provides suggestive evidence of a link between pattern completion in the hippocampus and predictive coding in early visual cortex (EVC)[3]. For pre-learned cue–action–outcome sequences, qualitatively different stimulus representations were evoked in the hippocampus and EVC for predictive actions (i.e., actions that determine an outcome given a cue). Given a part of a sequence, the hippocampus represented the full cue–action–outcome sequence, and this was related within and across participants to evidence of the same outcome in EVC. In control analyses with nonpredictive actions (i.e., actions that did not determine which outcome appeared after a cue), actions could not be decoded from stimulus-evoked activity in either region.

Beyond correlations in stimulus-evoked information, we hypothesize that the intrinsic coupling of the hippocampus and EVC may be enhanced during action-based prediction. This hypothesis is motivated by findings in human neurophysiology that link perceptual inference to long-range oscillatory synchronization between the hippocampus and visual cortex[6,7], together with the observation that stimulus-evoked responses and coherent spontaneous fluctuations are linearly superimposed in human functional magnetic resonance imaging (fMRI) data[8]. Critically, although correlated classification of stimulus-evoked responses is suggestive of hippocampal–neocortical interactions, such correlations depend upon the precision of memories and associated predictions represented within each region. Therefore, along with measuring multivariate patterns in the hippocampus and EVC, here we use background connectivity to quantify the temporal dynamics and covariance of these regions after removing stimulus-evoked responses[9,10]. Because background connectivity may more directly measure hippocampal–neocortical interactions than stimulus-specific decoding on its own, we reason that it should provide an objective index of the contexts in which the hippocampus is and is not involved in action-based predictive coding.

Critically, different accounts of memory retrieval make diverging predictions about the temporal contexts in which the hippocampus may be involved in predictive coding. On the one hand, involvement of the hippocampus could be specific to associations that have been learned very recently. In this case, hippocampal–neocortical interactions may diminish over time with the neocortex playing a more autonomous role in action-based prediction as a result of systems consolidation[11–13] (cf. ref. [14]). Alternatively, unique computational processes of the hippocampus like multimodal binding and pattern completion may serve an important function in prediction regardless of the more canonical role of the hippocampus as a memory system[15–17]. Thus, in the current study we test the role of the hippocampus in action-based prediction over two timescales. We hypothesize that background connectivity between the hippocampus and EVC depends on both the lag between training and scanning and the predictiveness of actions, and that this relates to the representational contents of these areas.

Participants learned cue–action–outcome sequences in a first training session 3 days before an fMRI scan and in a second training session immediately before the scan (Fig. 1a). Separate sets of cues and outcomes were used in each training session and actions were either predictive or nonpredictive of outcomes depending on the cue (Fig. 1b). For predictive actions, one outcome reliably followed the cue after a left button press and a different outcome reliably appeared after a right button press; explicit memory of predictable outcomes was at ceiling on verbal tests administered during each training session and before and after the fMRI scan (Fig. 1c). For nonpredictive actions, the two outcomes followed the cue with equal probability when either the left or right button was pressed. After both training sessions, participants performed the same task in the fMRI scanner, with stimuli from the two training sessions presented separately in alternating runs, and cues with predictive vs. nonpredictive actions presented separately in alternating blocks within each run type. Background connectivity was calculated for each of these blocks and then collapsed within condition, resulting in four key measures of hippocampal–EVC interaction: 3-day vs. no-delay learning of predictive vs. nonpredictive actions.

## Results

**Verbal tests**. To verify that predictive actions had been learned during training and remembered across the delay, participants were required to be 100% accurate in identifying expected outcomes of predictive actions in verbal outcome-identification tests outside of the fMRI scanner. Participants who did not reach this accuracy criterion on each test were excluded from the fMRI scan. Thus, by definition, all 24 scanned participants reached perfect accuracy. Two additional participants completed training but did not participate in the fMRI scan because of accuracy less than 100% even after repeating a pre-scan test.

**Choice RT**. Throughout training and in the scanner, we measured choice response time (RT) as the time it took for participants to press the left or right button in response to a cue. During training sessions outside of the scanner, choice RT did not differ among the conditions ($p$'s > 0.26 in repeated-measures ANOVAs). The lack of a timescale difference between training sessions is not surprising, as these conditions were equivalent at this point in the study. However, in the scanner, we observed a reliable interaction between timescale and predictiveness ($F(1, 22) = 5.49$, $p = 0.03$; Fig. 1d). For no-delay sequences, choice RT was comparable for predictive and nonpredictive actions ($t(23) = 0.18$, $p = 0.86$), whereas for 3-day delay sequences, choice RT was faster for predictive vs. nonpredictive actions ($t(23) = 3.96$, $p < 0.001$). When predictive and nonpredictive events were separately compared across delay conditions, speeded RT over time for predictive actions was marginally significant ($t(23) = 1.85$, $p = 0.08$), while slower RT over time for nonpredictive actions was not significant ($t(23) = 1.57$, $p = 0.13$).

**Stimulus-evoked responses**. A general linear model (GLM) containing finite impulse response (FIR) basis functions was used to estimate evoked blood-oxygen level-dependent (BOLD) activity in the hippocampus and EVC (Fig. 2a). Stimulus-evoked activity for each condition was averaged within block to capture the peak response. Although activity in the hippocampus was marginally reduced for both predictive and nonpredictive actions after the 3-day delay ($F(1, 22) = 4.09$, $p = 0.05$), no other main effects or interactions were observed in either ROI ($p$'s > 0.35 in repeated-measures ANOVAs).

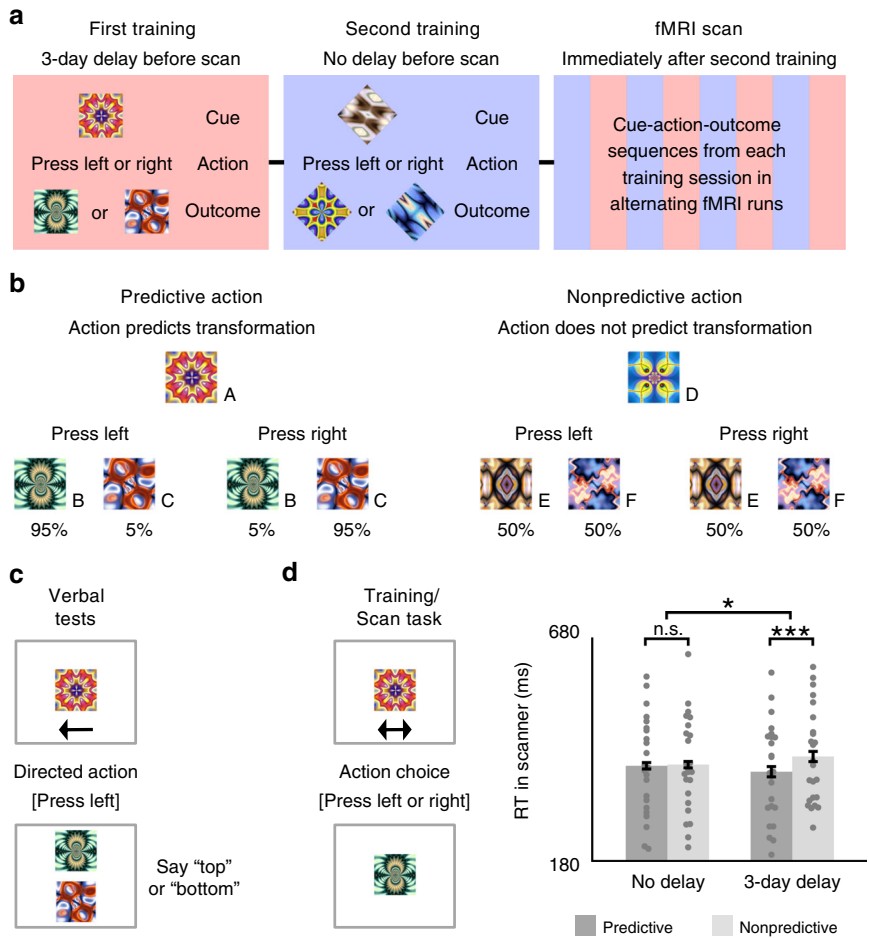

**Fig. 1** Design and RT. **a** The first training session was proctored 3 days before the fMRI scan, while the second training session was proctored immediately before the scan. To reduce interference, fractal stimuli in each session were masked to be either squares or diamonds in shape. The fMRI scan included four runs for each of the two training sets. **b** For predictive actions, given the cue, one outcome appeared with 95% probability when the left button was pressed and the other with 95% probability when the right button was pressed. For nonpredictive actions, given the cue, either outcome appeared with 50% probability when the left or right button was pressed. **c** In verbal tests outside the scanner, participants spoke aloud either "top" or "bottom" to indicate which outcome was expected given the cue and action. **d** For each trial in the fMRI scanner, participants chose to press either a left-hand or right-hand button to replace a cue with an outcome. While choice RT was similar for predictive and nonpredictive actions immediately after training, it was faster for predictive vs. nonpredictive actions after a 3-day delay. Error bars indicate ±1 SEM of the difference between predictive and nonpredictive actions at each timescale. ***$p < 0.001$; *$p < 0.05$ (paired $t$-tests). Source data are provided as a Source Data file

**ROI background connectivity**. Task-specific background connectivity between the hippocampus and EVC was measured after removing stimulus-evoked activity and confounding variables through linear regression in a multistep procedure[9,18–21]. We first used a GLM to regress out white matter and ventricle activity along with motion parameters from preprocessing, and then used FIR basis functions to capture and remove the average timing and shape of the hemodynamic response in each voxel in a data-driven way. Background connectivity was measured as correlations in the residual timeseries of each ROI. There were no differences across hemispheres in background connectivity between the hippocampus and EVC ($p$'s > 0.61 in repeated-measures ANOVAs). Critically, we observed a reliable interaction between timescale and predictiveness ($F(1, 23) = 8.28$, $p = 0.008$; Fig. 2b). This interaction was driven by a reliable difference between predictive and nonpredictive actions for sequences learned 3 days before the fMRI scan ($t(23) = 2.90$, $p = 0.008$), with no hint of an effect of predictiveness for sequences learned immediately before the scan ($t(23) = 0.12$, $p = 0.90$). When predictive and nonpredictive events were separately compared across delay conditions, enhanced background connectivity over time

for predictive actions was not significant ($t(23) = 1.67$, $p = 0.11$), while diminished background connectivity over time for nonpredictive actions was significant ($t(23) = 2.34$, $p = 0.03$). Furthermore, although the interactions between timescale and predictability in background connectivity paralleled interactions in RT, differences among conditions in background connectivity were not correlated with RT either across participants or across runs for each participant ($p$'s > 0.27 for all Pearson correlation coefficients and repeated-measures ANOVAs; Supplementary Fig. 1).

**Control correlations across matched runs**. The goal of background connectivity is to remove stimulus-evoked responses to isolate idiosyncratic fluctuations that reveal how experimental conditions modulate functional connectivity. To verify that the residualizing approach above was effective, we performed an across-run control analysis[9]. Each training condition was tested in two runs that used the same block order and the same cue–stimulus–outcome sequences. If the key findings above were confounded by unmodeled stimulus-evoked responses, the

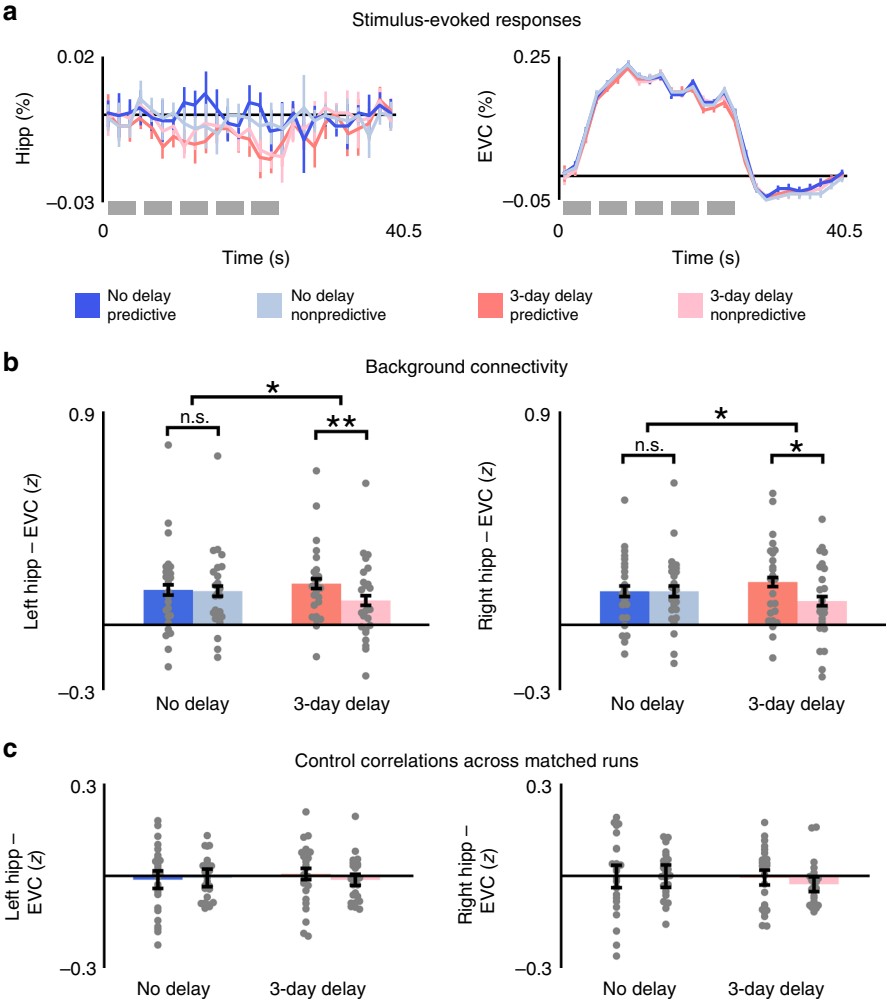

**Fig. 2** Stimulus-evoked responses and background connectivity. **a** Within each ROI, stimulus-evoked BOLD activity was similar during blocks of predictive and nonpredictive actions, both immediately after training and 3 days after training. Each block contained five trials (gray bars). **b** For both left and right hippocampus, background connectivity with EVC was stronger for predictive vs. nonpredictive actions 3 days after training but not immediately after training. **c** Control correlations across runs with matching stimuli and timing did not differ between any of the conditions. Error bars indicate ± 1 SEM of the difference between predictive and nonpredictive actions at each timescale. **p < 0.01; *p < 0.05 (paired t-tests). Source data are provided as a Source Data file

residual activity in the hippocampus in one run should be correlated with the residual activity in EVC in the other run. However, there were no reliable interactions or main effects when computing connectivity across runs ($p$'s > 0.17 in repeated-measures ANOVAs; Fig. 2c); moreover, connectivity was reliably lower for each condition when it was calculated across vs. within run ($p$'s < 0.01 in paired t-tests).

**Specificity within V1/V2.** Are differences in background connectivity specific to only the voxels that are most responsive to the specific retinotopic location of the experimental stimuli, or are they widespread throughout V1/V2? The EVC ROI for each participant included (1.5 mm isotropic) voxels responsive to square- and diamond-masked stimuli in a localizer scan, ranging from 732 to 4200 voxels in volume (6.2–30.0% of V1/V2). To examine the specificity of hippocampal background connectivity within EVC, we varied the extent of the EVC ROI from including just the 50 voxels (<1% of V1/V2) most responsive to functional localizer stimuli to including all V1/V2 voxels (Fig. 3a). Immediately after training, hippocampal background connectivity was equivalent for predictive and nonpredictive actions regardless of the size of the EVC ROI (Fig. 3b). In contrast, after the 3-day

delay, background connectivity was reliably stronger for predictive than nonpredictive actions across a wide range of ROI sizes (Fig. 3c). Likewise, the interaction between predictiveness and timescale was significant for ROIs ranging from 50 to 1000 voxels ($p$'s < 0.05) and marginally reliable for 5000 voxels ($F(1, 23) = 3.81$, $p = 0.06$). However, this interaction was not significant for the mean background timeseries across all V1 and V2 voxels ($F(1, 23) = 0.48$, $p = 0.50$). Thus, while differences in hippocampal background connectivity were robust to the size of the EVC ROI, they were not entirely pervasive within V1/V2.

**Voxelwise background connectivity.** To what extent are timescale and predictiveness differences in background connectivity between the hippocampus and EVC specific to these regions vs. widespread in the brain? To assess the anatomical specificity of the effects, we performed exploratory analyses using the residual timeseries from bilateral hippocampus and EVC ROIs (Fig. 4a) to calculate background connectivity with all voxels in the partial volume collected for each participant. After registering these correlation maps to MNI space, we conducted nonparametric randomization tests of their reliability across participants. Immediately after training, predictiveness did not reliably

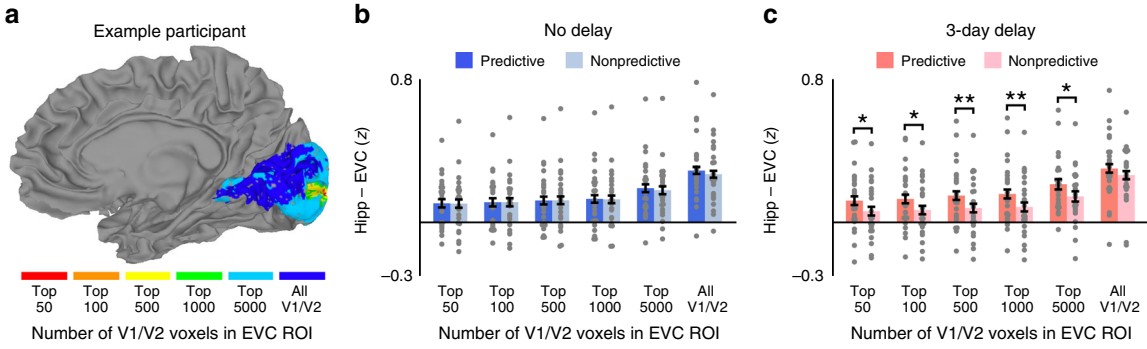

**Fig. 3 Specificity within V1/V2. a** The volume of the EVC ROI was incrementally dilated from including just the 50 voxels most responsive to the experimental stimuli in the functional localizer scan to including all of V1/V2. **b** Across the full range of ROI sizes, background connectivity was equivalent for predictive and nonpredictive actions after no delay. **c** After the 3-day delay, background connectivity was stronger for predictive than nonpredictive actions within ROIs that ranged in volume from 50 to 5000 voxels but was not reliably different when all V1/V2 voxels were included in the EVC ROI. Error bars indicate ± 1 SEM of the difference between predictive and nonpredictive actions at each timescale. **\*\***$p < 0.01$; **\***$p < 0.05$ (paired $t$-tests). Source data are provided as a Source Data file

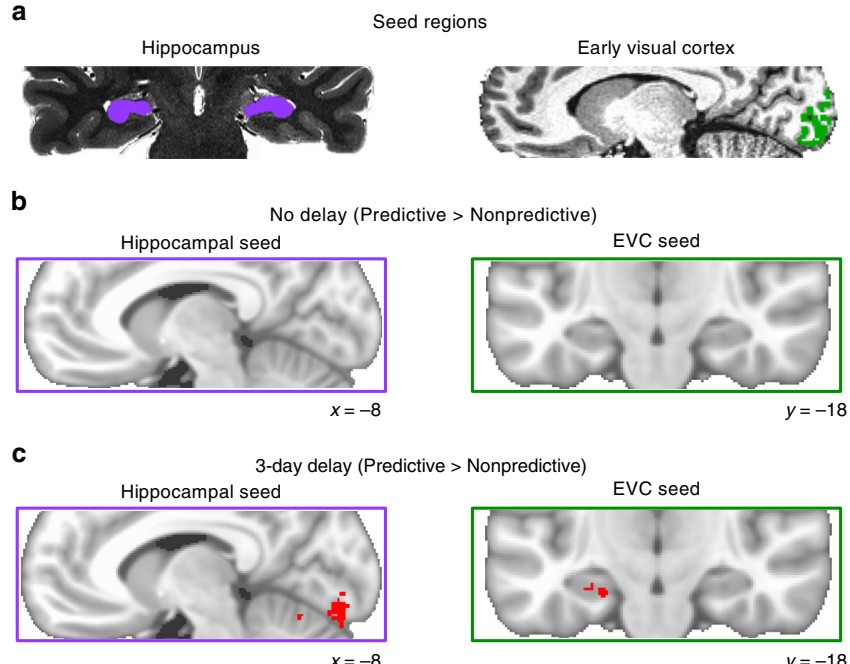

**Fig. 4 Voxelwise background connectivity. a** Hippocampal and EVC seed regions were defined on high-resolution anatomical scans. The hippocampal seed and ROI included CA2/3, dentate gyrus, and CA1 (but not the subiculum) since these subfields were linked to pattern completion during action-based prediction in a prior study with the same task[3]. EVC was limited to voxels in V1/V2 that were responsive to the experimental stimuli in a separate functional localizer. **b** Predictiveness did not reliably modulate hippocampal or EVC background connectivity anywhere in the field of view immediately after training. **c** However, we observed several clusters of enhanced background connectivity after the 3-day delay, including bilateral V1 and V2 for the hippocampal seed, and anterior and posterior left hippocampus for the EVC seed. Contrasts are visualized on the MNI152 template and corrected at $p < 0.05$ (TFCE) for the partial volume of functional scans outlined by purple/green boxes

modulate background connectivity anywhere in the partial volume when the hippocampus or EVC served as the seed (Fig. 4b). Conversely, for sequences learned 3 days before the scan, several clusters showed reliably greater background connectivity with each seed during blocks of predictive vs. nonpredictive actions (Fig. 4c). Specifically, predictiveness enhanced the background connectivity of the hippocampus with left (−9, −87, −13) and right (19, −91, −10) occipitotemporal cortex and left (−20, 6, −3) and right (25, 18, −4) putamen, and enhanced the background connectivity of EVC with anterior (−30, −12, −27) and posterior (−21, −42, −7) left hippocampus (bilateral at uncorrected threshold), left parahippocampal gyrus (−17, −52, 2), and

left posterior cingulate cortex (−9, −55, 14). At each timescale, no voxels showed stronger background connectivity with the hippocampus or EVC for nonpredictive actions.

**Verbal predictions for nonpredictive actions.** There are multiple potential explanations for the observed interaction between timescale and predictiveness in background connectivity. First, it could be that background connectivity between hippocampus and EVC was at equivalent baseline levels for predictive and nonpredictive actions immediately after training, while enhanced specifically for predictive actions

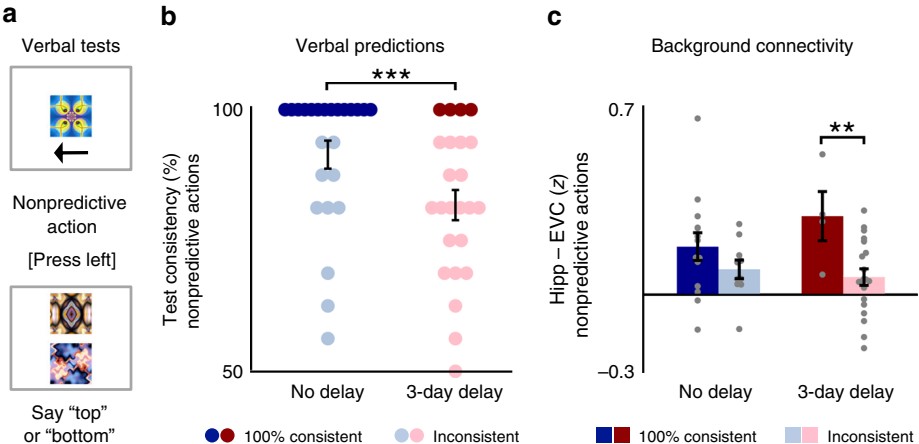

**Fig. 5** Verbal predictions for nonpredictive actions. **a** Verbal tests administered immediately before and after each fMRI scan included sequences with nonpredictive actions as well as sequences with predictive actions. **b** Participants more consistently identified particular outcomes for nonpredictive cues and actions learned immediately beforehand than before the 3-day delay. However, a subset of participants for each timescale were 100% consistent in identifying outcomes of nonpredictive actions. **c** Participants who consistently mapped nonpredictive cues and actions to particular outcomes showed greater background connectivity for these nonpredictive events after the 3-day delay. Error bars indicate ±1 SEM. ***$p < 0.001$ (paired $t$-test); **$p < 0.01$ (two-sample $t$-test). Source data are provided as a Source Data file

after the 3-day delay. Alternatively, it could be that background connectivity was already enhanced above baseline for both predictive and nonpredictive actions immediately after training, while reduced specifically for nonpredictive actions after the 3-day delay. Beyond the control correlations across matched runs that could be used to infer a baseline correlation for the context, behavior on the verbal tests before and after the fMRI scan can be used to help disentangle these possibilities. While participants were required to be 100% accurate in identifying outcomes of predictive actions, there were no correct or incorrect responses for nonpredictive actions. Nonetheless, participants could be consistent or inconsistent in their verbal predictions of unpredictable outcomes. We quantified this behavior for nonpredictive actions based on how consistently each participant mapped each outcome onto specific cue–action combinations (Fig. 5a). In fact, participants were significantly less consistent in verbally identifying expected outcomes of nonpredictive actions learned before the 3-day delay than for nonpredictive actions immediately before the scan ($t(23) = 3.86$, $p < 0.001$), suggesting that action-based prediction may have diminished over time for nonpredictive events (Fig. 5b).

Are consistent vs. inconsistent predictions sufficient to modulate hippocampal–neocortical interactions for nonpredictive actions? In total, 14 of the 24 participants were 100% consistent in identifying outcomes for nonpredictive cues and actions immediately after training, while 4 participants were 100% consistent after the 3-day delay. We reasoned that participants who were consistent in verbally identifying the outcomes of nonpredictive actions may have likewise maintained stronger visual predictions for nonpredictive actions than participants who were inconsistent in their responses. If so, such differences across participants may also be reflected in their hippocampal–neocortical interactions. Indeed, background connectivity during nonpredictive actions tended to be greater among participants who made 100% consistent test responses than among participants who made inconsistent responses (Fig. 5c). While this difference between participants was not significant immediately after training ($t(22) = 1.25$, $p = 0.22$), it was significant after the 3-day delay ($t(22) = 2.85$, $p = 0.009$). Moreover, among participants with 100% consistent test responses, background connectivity was the same for predictive and nonpredictive actions at each timescale ($p$'s $> 0.79$ in paired $t$-tests).

**Time-lagged background connectivity**. Background connectivity between the hippocampus and visual cortex during predictive action is agnostic to the direction of the interaction. Such questions can only be addressed definitively with techniques that allow for causal interventions. Moreover, the slow sampling rate of fMRI and the temporal autocorrelation of BOLD activity severely limit the analysis of temporal dynamics. Nevertheless, it is possible to test whether there exists any evidence for a temporal asymmetry in the signals between these regions[3] that would be consistent with processing in one region preceding the other. Specifically, we hypothesized that insofar as the hippocampus is relying on learned predictiveness to reinstate expected outcomes in visual cortex, the activity in the hippocampus at one time point should predict activity in visual cortex at the next time point, at least more than the reverse. Indeed, we were able to replicate the main timescale by predictiveness interaction reported above when EVC was lagged by one time point with respect to the hippocampus ($F(1, 22) = 4.77$, $p = 0.04$; Fig. 6a). This interaction reflected a reliable difference in background connectivity between predictive and nonpredictive actions for sequences learned three days before the fMRI scan ($t(23) = 2.98$, $p = 0.007$) and not for sequences learned immediately before the scan ($t(23) = -0.33$, $p = 0.74$). Critically, using nonpredictive blocks as the baseline controls for the possibility that BOLD activity merely peaks later in visual cortex than the hippocampus. In contrast, no such interaction was found when the hippocampus was lagged with respect to EVC ($F(1,22) = 0.04$, $p = 0.85$; Fig. 6b), with no differences between predictive and nonpredictive actions at either timescale ($p$'s $> 0.21$ in paired $t$-tests).

**Multivariate pattern similarity**. We have shown that background connectivity between the hippocampus and EVC strengthens over time for predictive relative to nonpredictive actions. How does this relate to the information represented in each ROI? Specifically, we hypothesized that greater connectivity for predictive actions after 3 days should be accompanied by greater information about expected outcomes. We tested this by measuring the neural similarity between visual transitions in which the same cue appeared but was followed by different outcomes (Fig. 7a). Insofar as these overlapping cue–outcome transitions are more differentiated after 3 days vs. immediately

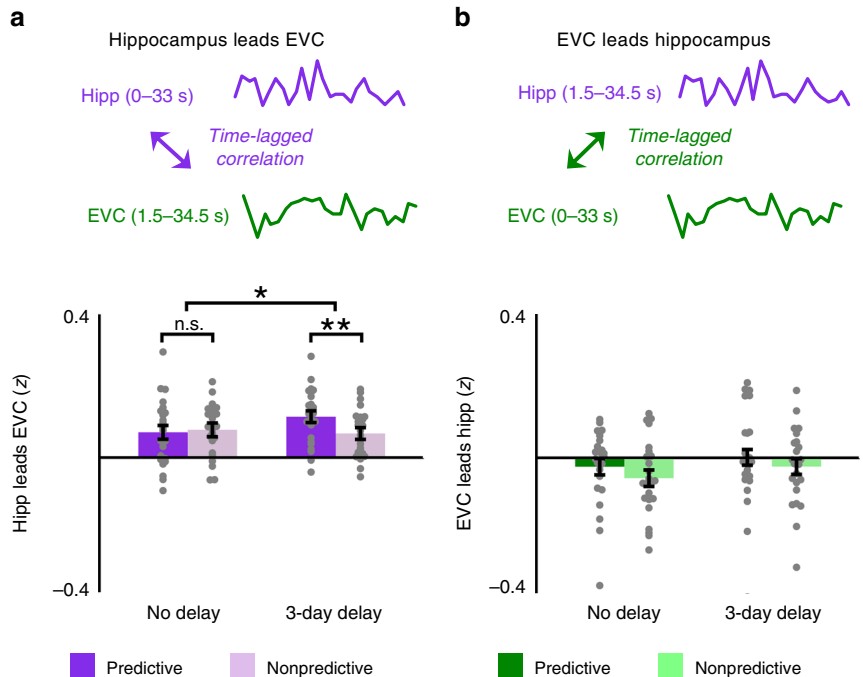

**Fig. 6** Time-lagged background connectivity. **a** Hippocampus leads EVC: For each block we computed the correlation between earlier background activity in the hippocampus (0–33 s) with later background activity in EVC (1.5–34.5 s); a similar pattern of results emerged as for non-shifted background connectivity. **b** EVC leads hippocampus: For each block we correlated earlier background activity in EVC (0–33 s) with later background activity in the hippocampus (1.5–34.5 s); there were no reliable differences among conditions. Error bars indicate ± 1 SEM of the difference between predictive and nonpredictive actions at each timescale. **p < 0.01; *p < 0.05 (paired t-tests). Source data are provided as a Source Data file

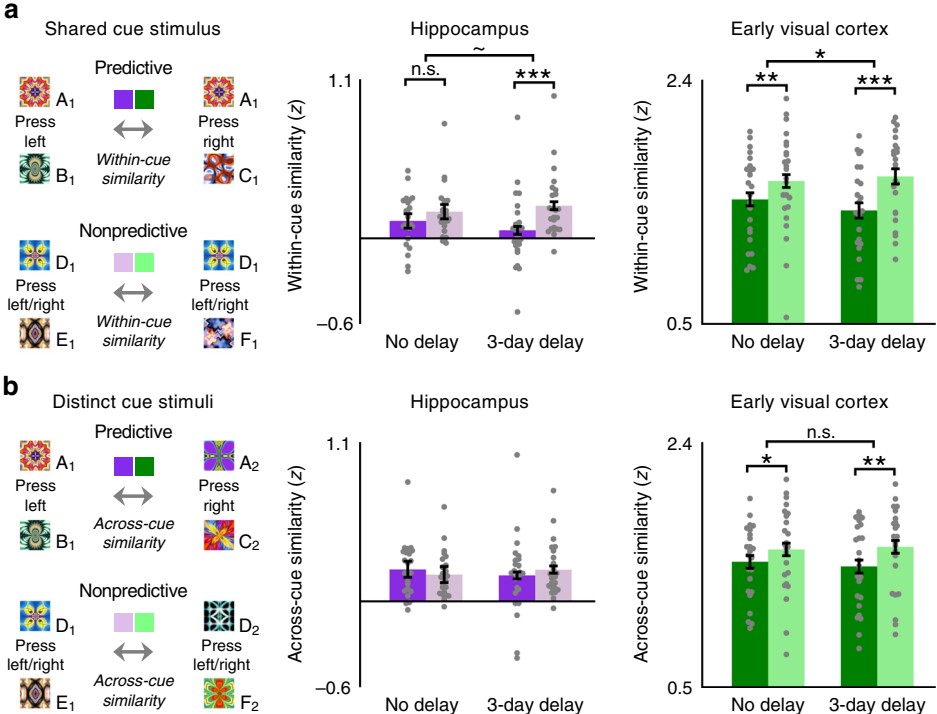

**Fig. 7** Multivariate pattern similarity. **a** Within-cue pattern similarity was measured as the correlation across voxels between cue–outcome visual transitions that shared the same cue but contained different outcomes. Hippocampal and EVC within-cue similarity were lower for predictive than nonpredictive actions after a 3-day delay, compared to no delay. **b** Across-cue pattern similarity was measured between cue–outcome transitions with non-overlapping stimuli. Delay interval did not modulate across-cue similarity in either the hippocampus or EVC. Error bars indicate ± 1 SEM of the difference between correlations for predictive and nonpredictive events at each timescale. ***p < 0.001; **p < 0.01; *p < 0.05; ~p = 0.051 (paired t-tests). Source data are provided as a Source Data file

after training, it would imply that the actions led to a stronger and/or clearer prediction of the outcome. To calculate pattern similarity, we correlated spatial patterns of parameter estimates in the hippocampus and EVC obtained from an event-related GLM, as a function of which cue was presented, whether it was associated with predictive vs. nonpredictive actions, and at what timescale it was learned. Critically, visual stimulation was the same for cue–outcome transitions containing either predictive or nonpredictive actions: either of two outcomes followed a cue and double-sided arrow with equal probability. But since nonpredictive actions could not be decoded in either the hippocampus or EVC in a prior study with the same action-based prediction task[3], we averaged across left and right button presses for visual transitions that contained nonpredictive actions. However, all of the pattern similarity effects replicated in follow-up analyses with resampled data that split between left and right nonpredictive actions (Supplementary Fig. 2). Moreover, predictiveness significantly interacted in each ROI with comparisons of within-cue vs. across-cue pattern similarity of the exact same multivoxel patterns ($p$'s < 0.01; Fig. 7b).

Consistent with prior findings[3], neural representations of the two alternative visual transitions associated with each predictive cue were less similar to one another than those of each nonpredictive cue in both the hippocampus ($F(1, 23) = 17.77$, $p < 0.001$) and EVC ($F(1, 23) = 32.16$, $p < 0.001$), suggesting that predictive actions helped disambiguate action outcomes in these regions. Importantly, differentiation effects were modulated by delay condition, including a marginally reliable interaction between predictiveness and timescale in the hippocampus ($F(1, 23) = 4.26$, $p = 0.051$) and a significant interaction in EVC ($F(1, 23) = 5.36$, $p = 0.03$). In the hippocampus, pattern similarity was reliably reduced for predictive vs. nonpredictive cues trained 3 days before the scan ($t(23) = 4.73$, $p < 0.001$), but did not differ for visual transitions trained immediately before the scan ($t(23) = 1.61$, $p = 0.12$). In EVC, despite a reliable interaction, the difference in pattern similarity for predictive and nonpredictive events was reliable both after the 3-day delay ($t(23) = 5.60$, $p < 0.001$) and immediately after training ($t(23) = 3.42$, $p = 0.002$). Unlike background connectivity, pattern similarity did not significantly differ between immediate and 3-day delay conditions within just predictive or just nonpredictive actions in either ROI ($p$'s > 0.10 in paired $t$-tests).

Do predictive events become more neurally distinct than nonpredictive events specifically when they share a cue (and thus initially overlap), or do they become more neurally distinct in general? To test whether differences in pattern similarity extend to non-overlapping events, we measured similarity between cue–outcome transitions with different cues (Fig. 7b). In the hippocampus, there was no difference in pattern similarity between predictive vs. nonpredictive events when both the cues and the outcomes were distinct ($F(1, 23) = 0.00$, $p = 0.97$). Conversely, this difference was reliable in EVC ($F(1, 23) = 17.24$, $p < 0.001$), with reduced similarity between cue–outcome visual transitions with predictive vs. nonpredictive actions. However, as noted above, within-cue vs. across-cue similarity reliably interacted with predictiveness in both the hippocampus ($F(1, 23) = 8.95$, $p = 0.007$) and EVC ($F(1, 23) = 11.33$, $p = 0.003$). Unlike the differentiation effect between overlapping cue–outcome transitions, across-cue similarity did not interact with delay condition in either the hippocampus ($F(1, 23) = 1.74$, $p = 0.20$) or EVC ($F(1, 23) = 0.96$, $p = 0.34$), though the three-way interaction of within-cue vs. across-cue similarity, predictiveness, and timescale was not reliable in either ROI ($p$'s > 0.12 in repeated-measures ANOVAs).

Finally, multivariate pattern similarity in the hippocampus was correlated across participants with background connectivity only after the 3-day delay. Individual differences across participants in background connectivity were unrelated immediately after training to within-cue pattern similarity in either the hippocampus ($r(22) = 0.11$, $p = 0.62$) or EVC ($r(22) = 0.10$, $p = 0.63$; Supplementary Fig. 3a). In contrast, after the 3-day delay, background connectivity was significantly negatively correlated with pattern similarity in the hippocampus ($r(22) = -0.62$, $p = 0.001$) though not EVC ($r(22) = -0.21$, $p = 0.32$; Supplementary Fig. 3b). Like background connectivity, pattern similarity in each ROI was not correlated with individual differences in RT at either timescale ($p$'s > 0.08 for all Pearson correlation coefficients). Since left and right button presses were averaged together in order to estimate multivoxel patterns corresponding to visual transitions, pattern similarity expectedly did not differ in either the hippocampus or EVC between participants who made 100% consistent vs. inconsistent verbal predictions for nonpredictive actions ($p$'s > 0.20 in two-sample $t$-tests).

## Discussion

Using high-resolution fMRI and a multi-session training paradigm, we examined how functional interactions between the hippocampus and EVC change over the early periods of a memory. Results build upon recent evidence of a link between hippocampal pattern completion and predictive coding in visual cortex[3], but suggest that the role of the hippocampus in visual prediction depends on the age of the knowledge on which the prediction was based. Specifically, interactions between the hippocampus and visual cortex became weaker for nonpredictive actions (and relatively stronger for predictive actions) 3 days after learning compared to immediately after learning. Over the same timescale, predictive actions led neural representations in these regions to become more differentiated for sequences with overlapping stimuli. Hippocampal prediction may be based at first on indiscriminate binding of co-occurring stimuli, with time and offline processing leading to gradual pruning of weaker associations, in this case, associations without informative actions.

Immediately after training, hippocampal–neocortical interactions were the same for predictive and nonpredictive actions. At first glance, the absence of a difference in background connectivity between these conditions may appear to be at odds with the finding that multivariate pattern similarity in EVC was significantly reduced for predictive vs. nonpredictive actions even immediately after training, and also with previous multivoxel pattern analysis (MVPA) findings in which classifier accuracy was at chance in both the hippocampus and EVC for nonpredictive actions while above chance for predictive actions[3]. Critically, however, background connectivity and MVPA are differentially sensitive to prediction in this task. Specifically, although participants cannot accurately predict outcomes of nonpredictive actions, they may nonetheless *inaccurately* predict outcomes. For example, the less predictable transitions for these cues may encourage hypothesis testing or other attempts to continue learning, or participants may be predicting both outcomes associated with the cue (which each still co-occur 50% of the time, far higher than any other outcome). Less differentiated patterns in visual cortex may in fact reflect less differentiated neural predictions, as opposed to a lack of prediction. Likewise, in any of these cases, a multivariate classifier will seek evidence of the correct outcome, and so performance will be at chance on average. However, to the extent that background connectivity between the hippocampus and visual cortex reflects the process of prediction, whether accurate or inaccurate, it may be enhanced for both predictive and nonpredictive actions.

Verbal predictions for nonpredictive actions before and after each fMRI scan—and their relationship across participants with

background connectivity—support the idea that the hippocampus may at first generate spurious predictions for nonpredictive events. While participants were required to be 100% accurate in identifying outcomes of predictive actions, there were no objectively correct or incorrect responses for nonpredictive actions. However, participants were more than 90% consistent on average in matching nonpredictive cues and actions to specific unpredictable outcomes immediately after training and were significantly less consistent in making such predictions for nonpredictive actions after the 3-day delay. Moreover, the small subset of participants who were still 100% consistent in their verbal predictions for nonpredictive actions after the 3-day delay exhibited significantly stronger background connectivity during nonpredictive events than did participants who made inconsistent predictions.

Three days after training, participants were also significantly quicker in making predictive actions than in making nonpredictive actions. Although RT in the scanner did not correlate with background connectivity across participants, quicker responses to predictive cues coincided across conditions with greater background connectivity. Accordingly, changes in hippocampal–neocortical interaction may relate to the perceptual fluency of cue–action–outcome sequences. At the same time that strengthening of sparse hippocampal representations may lead to faster responses to predictive cues, weak or noisy representations for nonpredictive associations may lead to slower responses to nonpredictive cues. While time-dependent changes in perceptual fluency may be independent of the hippocampus for tasks that are completely perceptual[22,23], hippocampal function is necessary for learning arbitrary associations among stimuli[24]. Notably, however, the statistical learning required for action-based prediction may involve different pathways within the hippocampus than other forms of hippocampally dependent learning[5,24].

While background connectivity effects were robust across a wide range of ROI sizes within EVC, voxelwise background connectivity largely overlapped with the specific a priori ROIs. Immediately after training, predictive actions did not reliably modulate voxelwise background connectivity with either the hippocampus or EVC. However, after a 3-day delay, predictive actions significantly modulated hippocampal background connectivity with voxels in V1 and V2, as well as EVC background connectivity with voxels in the hippocampus. In addition to overlap with the specific ROIs, a few other interesting findings emerged including enhanced hippocampal background connectivity for predictive actions with the putamen and object-selective visual areas in posterior fusiform and lateral occipital cortex. The putamen is especially intriguing because it has frequently been linked with action selection[25] and with offline motor-sequence learning[26]. Moreover, this finding converges with previous MVPA findings for action decoding in the putamen[3].

Along with background connectivity and behavior, multivariate pattern similarity within the hippocampus and visual cortex depended on the combination of predictiveness and delay interval. In the hippocampus, consistent with hippocampal models of episodic memory that emphasize the importance of representational overlap for neural differentiation[27,28], we observed reduced pattern similarity for predictive relative to nonpredictive actions only between visual transitions that shared the same cue stimulus. In EVC, predictive actions led to more distinctive neural patterns at each timescale for both overlapping visual transitions (that shared a cue stimulus) and non-overlapping transitions (in which both the cue and outcome differed). However, just as observed in the hippocampus, delay condition significantly modulated the effect of predictiveness on EVC pattern similarity only between overlapping visual transitions. Thus, the passage time modulated neural differentiation effects in EVC in the same way as in the hippocampus, further linking these regions together.

At the same time that changes across time in hippocampal–neocortical interaction are inconsistent with a time-invariant role for the hippocampus in predictive coding, models of memory retrieval that posit a reduced role for the hippocampus over time[11–13] would not obviously predict the findings: identical hippocampal–neocortical interaction during predictive and nonpredictive actions immediately after training followed by greater interaction specifically during predictive actions after a 3-day delay. In order to accommodate these findings, models that include the hippocampus need to include a role for predictive action in offline processing. Specifically, predictive action may provide a mechanism for prioritizing which representations are either strengthened through synaptic potentiation or weakened through synaptic depression during periods of offline rest[29,30]. Activity-dependent synaptic potentiation and depression may in turn be mediated by offline replay within the hippocampus[31,32] and between the hippocampus and neocortex[33,34]. By transforming noisy recent associations into sparser remote associations, this offline processing may increase the efficiency and utility of hippocampal associations over time[35,36]. Ultimately, sparser hippocampal representations may increase the signal-to-noise ratio of the hippocampal–neocortical interactions during action-based prediction.

While feedback across layers of visual cortex may be sufficient to fill-in adjacent elements of a sequence or scene, top-down connections such as from the hippocampus may be needed to simultaneously predict multiple elements in a sequence[37,38] and to make predictions based on prior co-occurrence and arbitrary associations[3,16]. Indeed, time-lagged background connectivity here converges with previous MVPA findings in which sequence information in the hippocampus temporally preceded outcome information in EVC during mnemonic prediction[3]. The timescale by predictiveness interaction observed for background connectivity was preserved when hippocampal background activity was shifted earlier to lead EVC, while it was eliminated when hippocampal background activity was shifted later to trail EVC. Although the causal direction of the relationship between the hippocampus and EVC cannot be established with correlational measures such as fMRI, converging data across this experiment and a previous study[3] are at least consistent with the hippocampus reinstating expected outcomes in visual cortex.

Hippocampal–neocortical interactions measured here through background connectivity are consistent with previous findings in human neurophysiology that link perceptual inference to the synchronization of long-range hippocampal-cortical oscillations[6,7]. Because stimulus-evoked responses and coherent spontaneous fluctuations are linearly superimposed in human fMRI data[8], intrinsic activity within the hippocampus and EVC can be separated from stimulus-evoked responses and other variables[9,10]. Whereas correlations in classification of stimulus-evoked responses depend upon the precision of memories and associated predictions represented within each region, background correlations may more directly reflect hippocampal–neocortical interactions themselves. In this way, background connectivity provides a more objective index of hippocampal involvement in action-based predictive coding. By using background connectivity to reveal consolidation-related effects on visual prediction, findings here further develop the link between hippocampal representation[2,39] and models of predictive coding in visual cortex[40,41].

In sum, interactions between the hippocampus and EVC, and representations in these areas, strengthen over time for predictive actions relative to nonpredictive actions. Hippocampal prediction

may occur by default, based at first on indiscriminate binding of co-occurring of stimuli. Time and offline processing may gradually prune weaker associations, in this case ones without informative actions, so that hippocampal reinstatement becomes increasingly specific to predictive events.

## Methods

**Participants**. Twenty-four individuals (19 female, aged 18–33 years) from the Princeton University community participated in the study. Each participant was right-handed and had normal or corrected-to-normal vision. Two additional participants completed the training sessions but did not participate in the fMRI component of the experiment due to below-criterion accuracy on verbal outcome-identification tests prior to the scan. Participants were paid $20 per hour and provided informed consent to a protocol approved by the Princeton University Institutional Review Board.

**Stimuli**. The primary stimulus set included 24 fractal-like images that were masked to be either square or diamond in shape. An additional 144 unique fractal and phase-scrambled images were included in a localizer to identify V1/V2 voxels reliably responsive to the experimental stimuli. All fractal images were created using ArtMatic Pro (www.artmatic.com). Both square- and diamond-masked stimuli subtended ~4° of visual angle in diameter on the training/testing laptop computer, and 4.5° in the scanner. We counterbalanced the assignment of images to 3-day delay and no-delay conditions and to sequences containing either predictive or nonpredictive actions, and randomly assigned images to serve as cues or outcomes. The Psychophysics Toolbox[42] for MATLAB (MathWorks) was used for stimulus presentation and response collection.

**First training session (3-day delay)**. The first training session was proctored 3 days before the fMRI scan on a laptop computer in a behavioral testing room. As in previous studies involving the same action-based training paradigm, the training session began with an exploratory training phase, followed by a verbal outcome-identification test, and finally a directed training phase[3,43]. The exploratory training phase included 320 trials in which a cue stimulus appeared on the computer screen for 1000 ms and then a double-headed arrow appeared below the cue. Participants were allowed an unlimited amount of time for each trial to make either a left button press or a right button press, in order to replace the cue with an outcome stimulus that appeared for 1000 ms. A meter at the bottom of the screen tracked the proportion of left and right button presses throughout the exploratory training phase, and participants were instructed to keep the meter pointer within a specified central zone, in order to roughly equate the frequency of actions and outcomes.

The directed training phase included 160 trials in which the onset of the cue was followed by a single-headed arrow that instructed participants to make a left or right button press for that trial. Directed training was included in order to equate the stimulus frequencies and transitional probabilities of the two outcomes associated with each cue throughout training. For example, if participants responded left more than right during the exploratory training, they were more likely to be instructed to respond right in the directed training.

**Second training session (no delay)**. The second training session was also proctored on a laptop computer in a behavioral testing room, but immediately before the fMRI scan and with new cue and outcome stimuli. To minimize interference between stimulus sets from different sessions, we masked one set with squares and the other with diamonds, with the order counterbalanced across participants. The structure of the second training session was identical to the first, with an exploratory training phase, then a verbal outcome-identification test, and finally a directed training phase.

**Predictive actions**. For half of the sequences within each training session, actions were highly predictive of the outcome. For instance, given the predictive cue A, outcome B appeared with 95% probability when the left button was pressed, and an outcome C appeared with just 5% probability. Similarly, when the right button was pressed, outcome C appeared with 95% probability and outcome B appeared with just 5% probability. Within each training session, participants were exposed to two different cue stimuli for which actions were highly predictive of outcomes.

**Nonpredictive actions**. Randomly intermixed with the predictive action trials, the remaining half of the sequences within each training session contained nonpredictive actions: the two outcomes for each cue appeared with equal probability, irrespective of which button was pressed. That is, given the nonpredictive cue D, outcome E or outcome F appeared with 50% probability when either the left or right button was pressed. Within each training session, participants were exposed to two different cue stimuli for which actions were nonpredictive of outcomes.

**Scan task**. The task in the fMRI scanner resembled the training sessions. Participants were instructed to continue to keep track of probabilistic relationships between button presses and fractal pairs while in the scanner and they knew to

expect a final set of behavioral tests after the scan. A total of 320 sequence trials were organized into eight 6-min runs. Each run contained sequences from either the first training session or the second training session, alternating between runs. Within each run, four blocks of predictive actions alternated with four blocks of nonpredictive actions. Pairs of runs for each participant contained the same stimuli and block order, while the trial order of the cue stimuli was randomized within and across blocks of predictive or nonpredictive actions. For nonpredictive actions, the trial order of the associated outcomes was also randomized within and across blocks. Each block included five trials and lasted 22.5 s, followed by 18 s of fixation. To match the outcome probabilities during the scan with the trained probabilities, participants were instructed to balance their left and right responses, and one block of predictive actions in each run contained a trial with an incorrectly predicted outcome (modeled separately and excluded from analysis).

As during exploratory training, each trial in the scanner involved three parts: a cue stimulus for 1000 ms, an action prompt consisting of a double-headed arrow below the cue that remained on screen until a button press or until the 1500 ms response window elapsed, and an outcome stimulus for 1000 ms. Participants used a separate response box for each hand to press the left and right buttons. If participants did not press a button within the response window, the cue stimulus and action prompt were replaced with a fixation cross that remained on screen until the next trial.

**Verbal tests**. Each participant performed six verbal tests over the course of the study: one test during each of the two training sessions (between the exploratory and directed training phases), one pre-scan test for each stimulus set directly before the fMRI scan, and one post-scan test for each stimulus set directly after the scan. On each test trial, a cue stimulus appeared at fixation. Below the cue, a single-headed arrow pointed either left or right, and participants were instructed to press the corresponding button. The cue and arrow disappeared, replaced by the two possible outcomes for that cue, presented above and below where the cue had been. For predictive actions, one outcome correctly completed the cue–action–outcome sequence, while the other outcome completed the cue–action–outcome sequence for the other action. Each verbal test included 16 trials (two trials for each cue–action–outcome sequence) with predictive and nonpredictive actions intermixed in a random order. Participants spoke aloud either "top" or "bottom" to indicate which outcome was expected. Verbal responses were used to avoid the button presses used for the trained actions. If a participant was less than 100% accurate for predictive actions in a verbal test, they were allowed to repeat the test one time without receiving feedback about which trials were answered incorrectly. Among the 24 scanned participants, seven participants repeated the pre-scan test for associations from the first training session, and one participant repeated the post-scan test for associations from the second training session. Two additional participants completed the training sessions but did not participate in the fMRI scan because their accuracy was less than 100% upon repeating a pre-scan test. Only the verbal tests that were administered immediately before and after the fMRI scan were used to calculate response consistency for nonpredictive actions.

**Choice RT**. Throughout training and in the scanner, we measured choice RT as the time it took for participants to press the left or right button in response to a cue. Although there was an unlimited amount of time for participants to respond during the exploratory and directed training phases, the response window was limited to 1500 ms after the action prompt in the scanner. To make the interpretation of choice RT comparable across the different parts of the experiment, we excluded trials from training in which choice RT exceeded a cutoff of 1500 ms.

**MRI acquisition**. Structural and functional MRI data were collected on a 3 T Siemens Skyra scanner with a 20-channel head coil. Structural data included a T1-weighted magnetization prepared rapid acquisition gradient-echo (MPRAGE) sequence (1 mm isotropic) for registration and segmentation of EVC, and two T2-weighted turbo spin-echo (TSE) sequences (0.44 × 0.44 × 1.5 mm) for hippocampal segmentation. Functional data consisted of T2*-weighted multi-band echo-planar imaging sequences with 42 oblique slices (16° transverse to coronal) acquired in an interleaved order (1500 ms repetition time (TR), 40 ms echo time, 1.5 mm isotropic voxels, 128 × 128 matrix, 192 mm field of view, 71° flip angle, acceleration factor 3, shift 2). These slices produced only a partial volume for each participant, parallel to the hippocampus and covering the temporal and occipital lobes. Collecting a partial volume instead of the full brain allowed us to maximize spatial and temporal resolution over our a priori ROIs. Data acquisition in each functional run began with 6 s of rest to approach steady-state magnetization. A B0 field map was collected at the end of the experiment.

**fMRI preprocessing**. Preprocessing was conducted using the Oxford Centre for Functional MRI of the Brain (FMRIB) Software Library 5.0 (FSL)[44]. Functional runs were corrected for slice-acquisition time and head motion, high-pass temporally filtered using a 128 s period cutoff, spatially smoothed using a 3 mm FWHM Gaussian kernel, and registered to each participant's high-resolution MPRAGE image using FLIRT boundary-based registration with B0-fieldmap correction[45].

**Hippocampal segmentation**. We anatomically defined hippocampal subfields on high-resolution $T_2$-weighted images for each participant, using the automatic segmentation of hippocampal subfields (ASHS) machine learning toolbox[46] and a database of manual medial temporal lobe segmentations from a separate set of 51 participants[47,48]. These manual segmentations were in turn based on anatomical landmarks from prior studies[49,50]. The hippocampal ROI was formed by combining CA2/3, dentate gyrus, and CA1. This was planned a priori because these subfields were linked to pattern completion during action-based prediction in our previous study[3].

**Early visual cortex**. The EVC ROI for each participant was anatomically constrained to V1 and V2, and functionally constrained to voxels reliably responsive to the experimental stimuli, as determined by an independent functional localizer. V1 and V2 were defined in each participant's T1-weighted anatomical scan using anatomical masks[51,52] generated with FreeSurfer[53]. During the functional localizer scan, participants detected one-back repetitions of fractals and scrambled images that had been masked to the same shapes (square, diamond) and sizes as the experimental stimuli. Fractal and scrambled stimuli were arranged into 16 blocks, each 15 s in duration with 9 s fixation between blocks. Within each block, 10 stimuli were each presented for 1000 ms followed by 500 ms fixation between trials. In total, the localizer run was ∼7 min in duration, and included 72 unique fractal images, along with 72 phase-scrambled versions of those images. Within the anatomical boundaries of V1 and V2, we selected voxels that were reliably responsive ($p < 0.05$) to both the square- and diamond-masked stimuli in the localizer. Defined in this way with 1.5-mm isotropic voxels, the volume of the EVC ROI ranged from 732 voxels to 4200 voxels (mean = 2699 voxels, SD = 847 voxels). Additionally, in order to examine the specificity of background connectivity within V1/V2, we incrementally varied the size of the EVC ROI from including all of V1/V2 (mean = 11,828 voxels, SD = 2464 voxels) to including only the 50 voxels in V1/V2 most responsive in an overall contrast of square- and diamond-masked stimuli in the localizer compared to baseline fixation.

**Stimulus-evoked activity**. A separate GLM containing FIR basis functions was applied to each run of the preprocessed data using FMRIB's Improved Linear Model (FILM)[44] with local autocorrelation correction. Each block and subsequent fixation period was modeled by 27 delta functions, one for each TR. Parameter estimates were averaged over TRs 5–19 of each block to capture the peak stimulus-evoked response. This included the stimulus duration shifted forward by 6 s in order to account for the hemodynamic lag. These stimulus-evoked responses were then averaged across voxels in each ROI and converted to percent signal change before combining across runs for each condition.

**ROI background connectivity**. Task-specific background connectivity between the hippocampus and EVC was measured after removing confounding variables and stimulus-evoked responses[9,18]. White matter and ventricle activity, along with six motion parameters, were regressed out of the preprocessed BOLD signal time-courses in a GLM for each run that was fit using FILM. Then, we estimated stimulus-evoked BOLD responses with the same FIR procedure above. Critically, the FIR basis functions captured the average timing and shape of the hemodynamic response in each voxel in a data-driven way. Within each run, we z-scored the residual (background) timeseries, extracted a 34.5 s time window of data for each block (from the start to 12 s after the last trial), and concatenated the data across blocks for each condition. Background connectivity was then measured as temporal Pearson correlations between concatenated background timeseries from each region and averaged across runs for each participant.

**Across-run control correlations**. Control analyses correlated background signal across pairs of runs containing the same stimuli and block order. Insofar as we successfully removed stimulus-evoked responses, across-run correlations should be eliminated, even when the same timeseries produce reliable correlations and correlation differences within run. The across-run correlations also quantify the contribution (if any) of stimulus-evoked responses to the residual background connectivity, which can be used as a baseline for within-run measures.

**Voxelwise background connectivity**. We performed exploratory analyses using the residual timeseries from bilateral hippocampal and EVC ROIs to calculate background connectivity with all voxels in the partial volume. The reliability of these maps was assessed across participants by registering the correlation maps for each seed ROI and condition to the MNI152 template space, which had been resampled with interpolation to match the resolution of the functional data (1.5 mm isotropic). Nonparametric randomization tests were performed for each voxel's connectivity using FSL Randomise[54], and corrected for multiple comparisons with threshold-free cluster enhancement (TFCE), resulting in a family-wise error rate of $p < 0.05$.

**Time-lagged background connectivity**. To examine the temporal dynamics of hippocampal-EVC background connectivity, we measured the temporal cross-correlation of background activity. Specifically, we shifted the time windows for

each block either forward or backward to assess temporal precedence. To test for evidence that the hippocampus leads EVC, we computed the within-block Pearson correlations between background signal in the hippocampus from 0 to 33 s and EVC from 1.5 to 34.5 s. Likewise, to test for evidence that EVC leads the hippocampus, we computed the within-block correlations between the hippocampal background activity from 1.5 to 34.5 s and EVC background activity from 0 to 33 s. To avoid concerns about the relative timing of the BOLD response between region, we are not interested in the overall magnitude of cross-correlations, but rather in modulation of these cross-correlations by experimental condition.

**Multivariate pattern similarity**. Multivoxel patterns in the hippocampus and EVC for each cue–outcome visual transition were based on parameter estimates of BOLD response amplitude in an event-related GLM for each run. Each cue–outcome transition was modeled with its own regressor and temporal derivative, constructed by convolving a boxcar function that matched the average trial duration for the condition (between 2188 and 2643 ms, depending on the participant's mean response time) with a double-gamma hemodynamic response function. This resulted in eight regressors of interest: four regressors for the cue–outcome transitions associated with predictive actions (e.g., A1-left-B1 and A1-right-C1), and four regressors for the cue–outcome transitions associated with nonpredictive actions (e.g., D1-[left/right]-E1 and D1-[left/right]-F1). Each GLM was fit using FILM with local autocorrelation correction and six motion parameters as nuisance covariates, as well as an additional regressor and its temporal derivative to model the single predictive event within each run that contained a counter-predicted outcome, along with trials for which the participant failed to press a button before the response deadline. Parameter estimates for each visual transition were then averaged across runs before calculating pattern similarity.

Pattern similarity was measured as z-transformed Pearson correlations across voxels within each ROI. Within-cue similarity was measured as the correlation between cue–outcome transitions containing the same cue but different outcomes (e.g., A1-left-B1 vs. A1-right-C1 for predictive actions and D1-[left/right]-E1 vs. D1-[left/right]-F1 for nonpredictive actions). Across-cue pattern similarity was measured as the correlation between cue–outcome transitions containing completely distinct cue and outcome stimuli (e.g., A1-left-B1 vs. A2-right-B2 for predictive actions and D1-[left/right]-E1 vs. D2-[left/right]-F2 for nonpredictive actions). For cue–outcome transitions with predictive actions, across-cue pattern similarity was measured across left and right button presses in the same way as within-cue pattern similarity.

Cue–outcome transitions were visually identical for predictive and nonpredictive actions—either of two outcomes followed a cue stimulus and a double-sided arrow with equal probability as one another. However, while button presses differed across alternative visual transitions for each predictive cue, this was not the case for nonpredictive cues (in which either button press could produce either outcome). Since nonpredictive actions could not be decoded in either the hippocampus or EVC during action-based prediction in a previous study with the same task paradigm[3], we averaged across left and right button presses in order to estimate the multivoxel patterns for visual transitions with nonpredictive actions. Importantly, averaging in this way balanced the number of observations used to estimate neural patterns for each condition, thereby equating the contrast-to-noise ratio (CNR)[55] of patterns for each condition (Supplementary Fig. 2b). Equating CNR across conditions is important for comparing pattern similarity across conditions because voxel-level variability strongly influences multivoxel correlations among patterns[56,57].

As a follow-up control analysis to ensure that averaging across left and right button presses for visual transitions with nonpredictive actions did not bias the primary findings, we additionally calculated the within-cue pattern similarity between visual transitions with left vs. right button presses for nonpredictive actions (Supplementary Fig. 2). For this analysis, each nonpredictive cue–action–outcome sequence was modeled with its own regressor and temporal derivative (e.g., D1-left-E1, D1-left-F1, D1-right-E1, and D1-right-F1). Within-cue similarity was then measured specifically between cue–action–outcome sequences that shared the same cue but differed in both the action and the outcome (e.g., D1-left-E1 vs. D1-right-F1 and D1-left-F1 vs. D1-right-E1). Notably, splitting up trials in this way for nonpredictive actions cut in half the number of trials for estimating the neural pattern for each cue–outcome visual transition (from about 5 trials per run in the primary analyses to about 2.5 trials per run in the resampled analysis). In order to calculate pattern similarity in the same way for visual transitions containing either predictive or nonpredictive actions (and thereby equate CNR across conditions), we randomly resampled trials with predictive actions as belonging to either of two partitions ('a' or 'b'). Then, we calculated within-cue pattern similarity between the randomly sampled dataset (e.g., A1a-left-B1a vs. A1a-right-C1a and A1b-left-B1b vs. A1b-right-C1b) before averaging together correlation coefficients in the same way as for visual transitions containing nonpredictive actions.

**Statistics**. All correlation coefficients calculated for background connectivity and pattern similarity were Fisher z-transformed prior to statistical analysis. In ROI analyses, pattern similarity and background connectivity were calculated separately for each hippocampal hemisphere and then averaged across hemispheres to reduce multiple comparisons. Repeated-measures ANOVAs and paired-sample t-tests

were used to compare RT, test consistency, background connectivity, and pattern similarity for predictive and nonpredictive actions. Pearson correlations were used to compare the between-subject variability in RT, background connectivity, and pattern similarity. Two-sample t-tests were used to compare between groups of participants who were either consistent or inconsistent in their verbal predictions for nonpredictive actions. All tests were evaluated against a two-tailed $p < 0.05$ level of significance.

**Reporting Summary.** Further information on research design is available in the Nature Research Reporting Summary linked to this article.

## Data availability

A reporting summary for this Article is available as a Supplementary Information file. The source data underlying Figs 1d, 2a–c, 3b–c, 5b–c, 6a–b, and 7a–b and Supplementary Figs 1a–b, 2b–c, and 3a–b is provided as a Source Data file. All neuroimaging data and experimental stimuli are freely available through the OpenNeuro platform for sharing neuroscience data (OpenNeuro.org) with DOI 10.18112/openneuro.ds001946.v1.0.0 [https://openneuro.org/datasets/ds001946/versions/1.0.0].

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

## Acknowledgements

This work was supported by NIH grants F32 EY023162, R01 EY021755, and R01 MH069456.

## Author contributions

N.C.H., E.W.A. and N.B.T.-B. reviewed the analyses, discussed the results, and wrote the paper. N.C.H. and N.B.T.-B. designed the experiment. N.C.H. and E.W.A. collected the data and performed the analyses.

## Additional information

**Competing interests:** The authors declare no competing interests.

