## [Peer Review File · Nature Communications]

Reviewers' Comments:

Reviewer #1:

Remarks to the Author:

The manuscript focuses on interactions between hippocampus and visual cortex, and how these change over time after participants learn to associate actions with outcomes. Some of the actions reliably predicted their outcome, while other actions did not. Immediately after learning these associations, connectivity between hippocampus and visual cortex was similar for predictive and non-predictive actions. However, three days after learning, predictive action-outcome associations led to stronger hippocampal-neocortical interactions than non-predictive actions. The authors furthermore report that in visual cortex the neural representations of predictive outcomes are more distinct than the representations of non-predictive outcomes.

The manuscript is very well written and the work appears mostly methodologically sound (but see below). The results are novel and timely. The finding that hippocampus-neocortical interactions sharpen over time for predictive (but not non-predictive) outcomes is interesting, and suggests that predictiveness plays a role in consolidation-related processes. My enthusiasm is dampened, however, by a potential confound in one of the analyses that makes it difficult to interpret some of the results, as explained below.

- Are visual stimuli more distinct for predictive versus non-predictive actions?

The authors find that neural representations of predictable outcomes are less similar to one another than those of non-predictable outcomes, but I wonder to what degree this effect is driven by the visual stimuli, rather than predictability per se. Consider, for example, two outcomes (a and b) and two sequences (1 and 2). In the predictable condition, outcome a will be more likely in sequence 1, and outcome b in sequence 2. In the non-predictable condition, however, either outcome will be equally likely for sequence 1 and 2. To address neural similarity, the authors calculated the correlation across voxels for sequences that shared the same cue but contained different outcomes. As far as I understand their analysis, this would be a correlation between sequence 1 and 2 in the example above. However, this analysis would automatically result in reduced similarity for the predictable condition, even when there was no neural effect due to predictability, simply because of the way the sequences were constructed (i.e., across trials, sequences 1&2 share fewer visual stimuli/outcomes in the predictable condition). I wonder to what degree this also holds for the actual sequences used in the experiment. One way to address this concern would be to perform a correlation analysis on the visual stimuli/sequences themselves to determine the amount of overlap in visual stimulation, both when they do and don't share a cue.

- A direct link between neural and behavioral effects?

It would strengthen the results if the neural effects predicted behavior. Does the strength of hippocampal-neocortical interactions predict behavioral (reaction time) effects, across runs and/or participants?

- Is enhanced connectivity between hippocampus and visual cortex specific to the retinotopic location of the stimulus?

It would be interesting to see the degree of overlap between voxels that respond to the retinotopic location of the stimuli (as identified by the visual localizer stimulus) and voxels that indicate enhanced connectivity with hippocampus (i.e. visual voxels identified in the partial volume analyses). One way to address this would be to visualize the degree of overlap on the inflated cortical surface.

Reviewer #2:

Remarks to the Author:

The manuscript by Hindy and colleagues uses human fMRI to probe interactions between hippocampus and early visual cortex during a cue-action-outcome learning task. Prior to fMRI scanning, subjects completed two training sessions (one just before scanning, one 3 days prior to scanning) during which they repeatedly saw various fractals and had to choose an action (left or right response) that led to an outcome. For half of the fractals, responses were highly predictive of outcomes (95% predictive) whereas for the other half of fractals, responses were non-predictive of outcomes (50% predictive). Thus, the two critical factors were whether a given cue was a predictive vs. non-predictive cue and then whether the cue was learned 3 days prior or just prior (same day) to scanning. During scanning, subjects essentially completed the same task with alternating blocks of predictive vs. non-predictive cues, and 3-day vs. same-day cues.

Behaviorally, subjects' reaction times in selecting actions did not differ for predictive vs. non-predictive cues learned the same day as scanning. For cues from the 3-day condition, however, subjects were markedly faster to make decisions in the predictive condition compared to the non-predictive condition. To be clear, this does not mean they were 'better'—simply that they spent less time deliberating about which response to make and, by extension, which outcome to 'reveal.' The primary fMRI measure that is used is background connectivity, which is a measure of the correlation between BOLD responses in different ROIs over time, after regressing out task-related activity. The idea is that this background connectivity is a measure of communication between regions that is not explained by task-evoked activity. Interestingly, background connectivity between the hippocampus and early visual cortex showed an interaction between delay and predictiveness. For predictive cues, connectivity tended to be stronger for the 3-day cues whereas for non-predictive cues, connectivity tended to be weaker for the 3-day cues. In other words, a longer delay seemed to increase connectivity for predictive cues but weaken connectivity for non-predictive cues. Another potentially interesting finding was that pattern similarity between shared cues with different outcomes tended to decrease in the 3 day condition (compared to the same day condition) but only for predictive cues. This was somewhat true in hippocampus, and in EVC. Interestingly, however, at least for the hippocampus, pattern similarity among non-shared cues showed a qualitatively different pattern with no evidence of a decrease in similarity for the 3 day condition.

While the manuscript addresses an interesting topic (delay-dependent changes in hippocampal-cortical interactions), I have several concerns that limit my enthusiasm. First, the results are discussed in terms of hippocampal predictions, but there is remarkably little in the results/analyses to compel or motivate a "prediction" account. I realize this interpretation is related to a previous, related paper, but the current study just doesn't seem to have any clear evidence that predictions are even occurring. Second, for several of the analyses, I have questions and/or concerns about how the analyses were performed and whether the interpretations are appropriate.

Comments:

1. The idea of using background connectivity to measure "predictions" that are generated by the hippocampus and then sent to EVC is not particularly intuitive to me. I can understand why we would expect predictions to be generated during the actual trials, but why would predictions also be going on "in the background?" Put another way, the fact that global BOLD signal in hippocampus is correlated with global BOLD signal in early visual cortex AFTER regressing out the task effects does not seem like an obvious marker of predictions being sent between these regions. Indeed, prior work from these

authors has used much more direct measures of prediction (e.g., by measuring reinstatement of predicted activity patterns). No clear rationale is provided as to why background connectivity is the right measure for the current question.

2. The changes in background connectivity strongly parallel the differences in reaction times across conditions—to a degree that raises serious concerns about what the background connectivity is actually reflecting. Specifically, this raises the obvious concern that the background connectivity measure is affected by reaction time. I'm not exactly sure how to test this with the current data, but it seems like a potentially serious issue. To be clear, I think the behavioral data are interesting, but given that the background connectivity approach involves regressing out task effects, and the task effects are based on trials with very substantial reaction time differences, it seems entirely plausible that the residuals would be influenced by the length of the trial that is being regressed out. Thus, it is hard to imagine that these RT differences would NOT influence the connectivity measures. If the RT differences do drive the differences in background connectivity, then it is obviously not appropriate to interpret the background connectivity as a measure of 'communication' between regions; rather it could just be a statistical artifact of regressing out task-related activity when the task conditions differ in reaction times.

3. Was Distinct cue similarity always based on cues that had different button presses (i.e., left vs. right). If not, then the relatively greater similarity among distinct cues could obviously be an artifact of the fact that the Different cue similarity includes trials with a common motor response whereas the Shared cue trials were necessarily restricted to trials that had opposite motor responses.

4. The pattern similarity results are interesting, but not overwhelmingly compelling. For one, the hippocampal interaction for shared cue data is only marginally significant. Additionally, for EVC, although there is a significant interaction between delay and predictiveness for the within-cue data and not for the across-cue data, it seems extremely unlikely that the 3-way interaction between delay, predictiveness and within-cue/across-cue is significant. For hippocampus, this three-way interaction might well be significant, though I don't think it is reported. But EVC certainly seems to show a very similar pattern for the within-cue and the across-cue analyses.

5. I was not fully clear on how the pattern similarity analyses were performed. Was there a separate parameter estimate for each trial in each run? Were correlations ever obtained between data from the same run, or was this restricted to across-run correlations? If same-run analyses were performed, are there any differences between the shared-cue and across-cue analyses in terms of lag effects? For example, depending on whether or not the trial order included consecutive repeats of the same condition, there might be a bias such that across-cue analyses were systematically based on trials that were further apart (in terms of mean trial lag), which could artificially make the across-cue similarity values increase. In any case, more detail is required about these analyses.

6. There is an interesting idea raised in the Discussion that in the no delay condition, the hippocampus might still be generating predictions, but they are as likely to be correct as incorrect which is why there is no difference for the predictive vs. non-predictive conditions. Yet, this argument seems to be contradicted by the behavioral data in that subjects were, in fact, required to reach 100% accuracy in a test of prediction memory before entering the scanner. Obviously, there is a change in behavior over time (reflected in RTs) and I do think that change is interesting, but given that accuracy was forced to be at ceiling, that seems to argue against the idea that subjects are generating inaccurate predictions in the no delay condition.

Reviewer #3:

Remarks to the Author:

In this manuscript, Hindy et al. present an intriguing follow-up to their prior work on a hippocampal role in predictive coding in visual cortex. They find that predictive cue-action-outcome sequences learned 3 days before fMRI scanning are associated with faster responses, greater state-based connectivity between hippocampus (HPC) and early visual cortex (EVC), and more dissimilar neural patterns in HPC and EVC versus sequences learned on the same days as scanning. Both the predictiveness of the sequences and the 3-day delay in learning had significant impacts on all measures. The authors argue that these findings point to a specific role for the HPC in binding cue-action-outcomes that exhibit regularity and that interactions between HPC and EVC for these predictive events strengthen over time.

Overall, this work is timely, well-conducted, and novel. The findings coalesce into a consistent package with fairly straightforward theoretical implications. This work will undoubtedly impact the field and motivate new research into the HPC as one of the brain's engines for prediction. There are, however, several issues detailed below that should be fully considered.

Major concerns

1) After reading the introduction, I already had the overall gist of my review in mind: too incremental. However, after reading the manuscript in its entirety, it is clearly much more than a simple follow-up to the authors' prior work and represents a novel contribution in its own right. As such, I think the introduction does not appropriately set the stage for the research. Although the authors have attempted to answer big important questions, the introduction has a much too narrow focus on their Nature Neuroscience paper. As is, the intro is not necessarily wrong, but it fails to motivate the larger question of hippocampal-cortical interactions for predictive coding to the more general audience of Nature Communications readers.

2) The authors motivate the study with two competing accounts of HPC's involvement in predictive coding: 1) HPC is endowed with the computations for prediction, thus should always be engaged in learning regularities and 2) HPC's role in prediction is restricted to early learning before consolidation at which point the baton is passed to cortical processes. Reasonably, they then run a task with manipulations of time thereby allowing for a test of two accounts. However, their findings do not seem to align well with either of the accounts. HPC processes and representations are not involved immediately after learning (thus ruling out account 1), but are involved after a 3 day delay (providing support against account 2). Although these findings are consistent with the authors' non-specific predictions that HPC-EVC interactions would depend on lag and predictiveness, they do not support either of the competing accounts. What updates in theory are needed to support the current findings? The authors provide hints of this in the discussion, but an explicit appreciation of this divergence from more conventional views of HPC function should be expanded.

3) Why background connectivity? Don't worry, I'm a fan of background connectivity and think the approach is underappreciated in the field. But, very little is provided to justify why background connectivity is a good measure for this study. More importantly, what do the connectivity findings imply at a mechanistic level, especially in light of their previous findings? I think the preferred argument is that this coupled activity arises due to HPC-guided cortical reinstatement. A more thorough motivation for using background connectivity and speculation for why functional coupling is mechanistically important for predictive coding would strengthen the manuscript and potentially encourage wider adoption of such connectivity methods.

4) At first glance, the difference in choice RT for predictive vs. nonpredictive sequences at a delay

provides converging evidence for the authors' claims. However, I hesitate on understanding what is driving this effect. Although there appears to be a bit of speeding for the delay predictive sequences, relative to no delay, the majority of this effect seems to be in slower responses to the delay nonpredictive sequences. Given that there are no "wrong" responses, what is slowing RTs to the nonpredictive sequences? In the discussion, the authors do describe an account of hippocampal-based strengthening of sparse representations for predictive associations, which is line with relatively faster responses to predictive sequences. But, what makes delayed nonpredictive slower relative to no delay? It may be that loosely-bound, weak representations of each nonpredictive cue-action-outcome sequence are all retrieved and compete for action selection. Also, I'm interested to see if individual differences in this behavioural effect are related to neural measures, either differences in connectivity or representational similarity (e.g., are faster RTs correlated with greater pattern dissimilarity for predictive sequences?). In any case, I do think the RT effect is interesting and may shed light on the mechanisms at play, but would appreciate the authors giving this effect more consideration in the discussion.

5) Pattern dissimilarity in EVC immediate suggests separation is occurring without the hippocampus. However, whereas the HPC results for the shared cue sequences are clearly different than the distinct cue sequences, EVC shows the same pattern for both. Two questions about the GLMs for estimating neural patterns in this analysis: 1) Were the action arrows included as part of the pattern-defining regressors (i.e., was the entire trial sequence modelled or just the cue? 2) Were all combinations of action pairs (left-right, left-left, right-right) included in the nonpredictive similarity comparisons? If the answer to both of these is yes, it may be that nonpredictive similarity is biased higher due to the matched visual stimulus from the action arrow (i.e., left-left and right-right trial pairs). The predictive similarity comparisons were always different actions, thus had less objective visual similarity across the sequence.

Assuming this confound isn't present or impacting the presented results, an extended discussion of these EVC pattern dissimilarity findings seems warranted. Is a different non-hippocampal top-down signal separating EVC patterns early in learning? Or is this an intrinsic computation of EVC?

Minor comments

Subiculum was not included in the hippocampal ROI. This is fine, as the authors' hypotheses about which subfields should matter are motivated by their prior work. However, the subiculum is a large portion of the hippocampus and its exclusion should be noted somewhere earlier in the paper than the methods.

pg. 11: In reporting stats for the interaction for the cross correlation between EVC and lagged HPC time series, a t stat is reported that should likely be a F stat.

It is unclear which part of each trial was included in the GLM parameters estimates for the multivariate patterns used in the representational similarity analysis. Were patterns based on models of the cue onset and duration or were the action and outcome also included in the model?

Responses to Editor

Your manuscript entitled "Hippocampal-neocortical interactions sharpen over time for predictive actions" has now been seen by 3 referees, whose comments are appended below. You will see that while the reviewers find your work of interest, they raised substantial concerns that must be addressed. In light of these comments, we cannot accept the manuscript for publication, but would be interested in considering a revised version that addresses these serious concerns.

We are pleased with the excitement about the potential significance of our study and very much appreciate the thoughtful comments, ideas, and suggestions from all three reviewers. We are confident that we have resolved all concerns raised by the reviewers and are grateful that the suggested revisions have substantially improved the manuscript.

In particular, the reviewers raise several doubts related to the strength of your results and interpretations. Reviewer 1 notes a confound, and Reviewers 2 and 3 question the appropriateness of your background connectivity measure, its influence on your interpretations, and the extent to which your results support the proposed account. Each of the reviewers provide suggestions for additional analyses that would strengthen the conclusions that can be drawn, which we strongly recommend you follow.

In order to address each reviewer's concerns, the revised manuscript includes two new subsections in the Results, each new subsection with its own new figure, along with three new supplementary figures. Additionally, we have thoroughly revised the Introduction and Discussion with respect to the rationale and interpretation of our background connectivity measure and have substantially expanded the Methods with respect to the pattern similarity analyses.

New findings from analyses suggested by the reviewers strengthen the results and support the interpretation that hippocampal-neocortical interactions initially reflect indiscriminate binding of co-occurring events and that nonpredictive associations are pruned over time through offline processing. In particular, Figure 5 ("Verbal predictions for nonpredictive actions") displays a newly discovered link between behavior and background connectivity. Figure 3 ("Specificity within V1/V2") reveals the robustness of the background connectivity effects within early visual cortex. Figure S1 ("RT vs. background connectivity") confirms that reaction time does not account for background connectivity effects. Figure S2 ("Resampled pattern similarity and contrast-to-noise ratios") confirms that button-press differences do not account for pattern similarity effects. Finally, Figure S3 ("Background connectivity vs. pattern similarity") reveals how background connectivity and pattern similarity are related across participants. Together, these new analyses substantially strengthen the body of evidence for our conclusions and we are very grateful for being prompted to include them.

Reviewer 3 also has suggestions for reframing the paper that we agree will increase the impact of your work.

We have followed Reviewer 3's thoughtful suggestions for reframing the paper in order to more directly focus on the novel contribution and larger significance of the ideas so that they will be obvious to the extremely broad audience of *Nature Communications*.

In your revision, please address these and the other concerns raised by the reviewers.

We thank the reviewers for their highly constructive feedback. We directly address each concern below.

Responses to Reviewer #1

The manuscript focuses on interactions between hippocampus and visual cortex, and how these change over time after participants learn to associate actions with outcomes. Some of the actions reliably predicted their outcome, while other actions did not. Immediately after learning these associations, connectivity between hippocampus and visual cortex was similar for predictive and non-predictive actions. However, three days after learning, predictive action-outcome associations led to stronger hippocampal-neocortical interactions than non-predictive actions. The authors furthermore report that in visual cortex the neural representations of predictive outcomes are more distinct than the representations of non-predictive outcomes.

The manuscript is very well written and the work appears mostly methodologically sound (but see below). The results are novel and timely. The finding that hippocampus-neocortical interactions sharpen over time for predictive (but not non-predictive) outcomes is interesting, and suggests that predictiveness plays a role in consolidation-related processes. My enthusiasm is dampened, however, by a potential confound in one of the analyses that makes it difficult to interpret some of the results, as explained below.

We are pleased that the reviewer found the study novel and timely. Additionally, we are indebted to the reviewer for their thoughtful and helpful feedback that prompted us to clarify the pattern similarity analyses, probe the retinotopic specificity of background connectivity within early visual cortex, and more carefully consider links between behavioral and neural effects.

- Are visual stimuli more distinct for predictive versus non-predictive actions?

The authors find that neural representations of predictable outcomes are less similar to one another than those of non-predictable outcomes, but I wonder to what degree this effect is driven by the visual stimuli, rather than predictability per se. Consider, for example, two outcomes (a and b) and two sequences (1 and 2). In the predictable condition, outcome a will be more likely in sequence 1, and outcome b in sequence 2. In the non-predictable condition, however, either outcome will be equally likely for sequence 1 and 2. To address neural similarity, the authors calculated the correlation across voxels for sequences that shared the same cue but contained different outcomes. As far as I understand their analysis, this would be a correlation between sequence 1 and 2 in the example above. However, this analysis would automatically result in reduced similarity for the predictable condition, even when there was no neural effect due to predictability, simply because of the way the sequences were constructed (i.e., across trials, sequences 1&2 share fewer visual stimuli/outcomes in the predictable condition).

We thank the reviewer for pointing this out and acknowledge that the original submission was ambiguous in its description of how patterns were constructed for multivariate similarity

analyses. In fact, comparisons between cue-outcome transitions were identical with respect to visual stimulation for predictive and nonpredictive events. That is, for both predictive and non-predictive actions, either of two outcomes followed a cue with equal probability. For instance, predictive cue “A” corresponded two possible cue-outcome (stimulus-stimulus) transitions: “A-left-B” and “A-right-C”. Likewise, nonpredictive cue “D” corresponded to two possible cue-outcome transitions: “D-[left/right]-E” and “D-[left/right]-F”. We have made revisions throughout the manuscript in order to clarify the pattern similarity analyses. In the Results, on page 17, we are now explicit that cue-outcome transitions were identically constructed with respect to visual stimuli for predictive vs. non-predictive actions. In the Methods, on page 35, we now explain in detail exactly how multivariate patterns were estimated and compared.

While visual stimulation was identical across conditions, we averaged across left and right button presses for cue-outcome visual transitions with nonpredictive actions but not predictive actions. Importantly, averaging in this way balanced the number of observations used to estimate neural patterns for each condition, thereby equating the contrast-to-noise ratio (CNR; Wald 2012, *Neuroimage*) of patterns for each condition (Figure S2B). Equating CNR across conditions is important because voxel-level variability strongly influences pattern similarity (Davis et al 2014, *Neuroimage*; Bhandari et al 2018, *JOCN*). In the revised manuscript, we have taken multiple measures outlined below in order to address any potential concerns about this approach.

First, we motivate averaging across nonpredictive left and right button presses based on a prior study with the same task in which nonpredictive actions could not be decoded in either the hippocampus or early visual cortex (Hindy et al., 2016 *Nature Neuroscience*).

Second, we note that—along with the reported interactions between predictiveness and timescale—predictiveness significantly interacted with comparisons of within-cue vs. across-cue pattern similarity in both the hippocampus and EVC ($ps < .01$; Results, page 18; Figure 7B). While interactions between predictiveness and timescale are interpretable on their own, interactions between predictiveness and within-cue/across-cue similarity are important for interpreting main effects of predictiveness on within-cue similarity. Because across-cue similarity was calculated on the same multivoxel patterns as within-cue similarity, differences between measures control for any differences in how we estimated the multivoxel patterns for predictive vs. nonpredictive events.

Finally, in Figure S2, all of the within-cue pattern similarity effects replicated in follow-up control analyses in which we additionally computed the within-cue similarity between left and right nonpredictive actions. For this analysis, we separately modeled each nonpredictive cue-action-outcome sequence, and measured within-cue similarity specifically between cue-action-outcome sequences that shared the same cue but differed in both the action and the outcome (e.g., “D-left-E vs. D-right-F” and “D-left-F vs. D-right-E”). Importantly, splitting up nonpredictive actions in this way cut in half the number of trials for estimating each neural pattern. Thus, in order to calculate within-cue similarity in the same way for predictive events, we randomly designated each predictive event as belonging to either the “Rand a” or “Rand b” dataset. This resampling equated the contrast-to-noise ratio among patterns for predictive and nonpredictive actions.

Results: Multivariate pattern similarity, p. 17, line 13:

“Critically, visual stimulation was the same for cue-outcome transitions containing either predictive or non-predictive actions: either of two outcomes followed a cue and double-sided arrow with equal probability. But since nonpredictive actions could not be decoded in either the hippocampus or EVC in a prior study with the same action-based prediction task³, we averaged across left and right button presses for visual transitions that contained nonpredictive actions. However, all of the pattern similarity effects replicated in follow-up analyses with resampled data that split between left and right nonpredictive actions (Fig. S2). Moreover, predictiveness significantly interacted in each ROI with comparisons of within-cue vs. across-cue pattern similarity of the exact same multivoxel patterns ($p < .01$; Fig. 7B).”

Results: Multivariate pattern similarity, p. 18, line 21:

“However, as noted above, within-cue vs. across-cue similarity reliably interacted with predictiveness in both the hippocampus $F(1, 23) = 8.95, p = .007$ and EVC $F(1, 23) = 11.33, p = .003$.”

Methods: Multivariate pattern similarity, p. 34, line 15:

“Each cue-outcome transition was modeled with its own regressor and temporal derivative, constructed by convolving a boxcar function that matched the average trial duration for the condition (between 2188 and 2643 ms, depending on the participant’s mean response time) with a double-gamma hemodynamic response function. This resulted in 8 regressors of interest: 4 regressors for the cue-outcome transitions associated with predictive actions (e.g., A1-left-B1 and A1-right-C1), and 4 regressors for the cue-outcome transitions associated with nonpredictive actions (e.g., D1-[left/right]-E1 and D1-[left/right]-F1).”

Methods: Multivariate pattern similarity, p. 35, line 12:

“Cue-outcome transitions were visually identical for predictive and nonpredictive actions – either of two outcomes stimuli followed a cue stimulus and a double-sided arrow with equal probability as one another. However, while button presses differed across alternative visual transitions for each predictive cue, this was not the case for nonpredictive cues (in which either button press could produce either outcome). Since nonpredictive actions could not be decoded in either the hippocampus or EVC during action-based prediction in a previous study with the same task paradigm³, we averaged across left and right button presses in order to estimate the multivoxel patterns for visual transitions with nonpredictive actions. Importantly, averaging in this way balanced the number of observations used to estimate neural patterns for each condition, thereby equating the contrast-to-noise ratio (CNR)⁵⁵ of patterns for each condition (Fig. S2B). Equating CNR across conditions is important for comparing pattern similarity across conditions because voxel-level variability strongly influences multivoxel correlations among patterns^{56,57}.

As a follow-up control analysis to ensure that averaging across left and right button presses for visual transitions with nonpredictive actions did not bias the primary findings, we additionally calculated the within-cue pattern similarity between visual transitions with left vs. right button presses for nonpredictive actions (Fig. S2). For this analysis, each nonpredictive cue-action-outcome sequence was modeled with its own regressor and temporal derivative (e.g., D1-left-E1, D1-left-F1, D1-right-E1, and D1-right-F1). Within-cue similarity was then measured specifically between cue-action-outcome sequences that shared the same cue but differed in both the action and the outcome (e.g., “D1-left-E1 vs. D1-right-F1” and “D1-left-F1 vs. D1-right-E1”). Notably, splitting up trials in this way for nonpredictive actions cut in half the number of trials for estimating the neural pattern for each cue-outcome visual transition

(from about 5 trials per run in the primary analyses to about 2.5 trials per run in the resampled analysis). In order to calculate pattern similarity in the same way for visual transitions containing either predictive or nonpredictive actions (and thereby equate CNR across conditions), we randomly resampled trials with predictive actions as belonging to either the “Rand a” or “Rand b” dataset. Then, we calculated within-cue pattern similarity separately for each randomly sampled dataset (e.g., “A1a-left-B1a vs. A1a-right-C1a” and “A1b-left-B1b vs. A1b-right-C1b”) before averaging together correlation coefficients in the same way as for visual transitions containing nonpredictive actions.”

Figure S2. Resampled pattern similarity and contrast-to-noise ratios. (A) In primary analyses of pattern similarity among cue-outcome visual transitions (Fig. 7), we averaged across left and right button presses for visual transitions with nonpredictive actions but not predictive actions. To ensure that averaging in this way did not bias the primary findings, here we calculated the within-cue similarity between visual transitions with left vs. right nonpredictive button presses. Additionally, in order to calculate within-cue similarity in the same way for visual transitions with predictive actions, we resampled trials with predictive actions as belonging to either the “Rand a” or “Rand b” dataset. (B) We calculated the contrast-to-noise ratio across voxels ($CNR = \sqrt{\bar{x}^2 / \sigma^2}$) of patterns entered into each analysis. Compared to the primary analyses, CNR of the resampled patterns was significantly lower in both the hippocampus ($F(1, 23) = 15.32, p < .001$) and EVC ($F(1, 23) = 57.75, p < .001$). Within each analysis, however, CNR did not differ between predictive and nonpredictive events ($ps > .36$). (C) Within-cue pattern similarity was lower overall among resampled patterns than in the primary analyses. However, differences in pattern similarity among the conditions were similar to those observed in the primary analysis of within-cue pattern similarity, including marginally reliable interactions between timescale and predictiveness in both the hippocampus ($F(1, 23) = 3.39, p = .08$) and EVC ($F(1, 23) = 2.97, p = .10$). In the hippocampus, within-cue similarity did not differ among

visual transitions trained immediately before the scan ($t(23) = 0.15, p = .88$), but was reliably lower for predictive events after a 3-day delay ($t(23) = 2.56, p = .18$). In EVC, within-cue similarity was marginally lower for predictive vs. nonpredictive events immediately after training ($t(23) = 1.78, p = .09$), and reliably lower after the 3-day delay ($t(23) = 4.53, p < .001$). Error bars indicate ± 1 SEM of the difference between predictive and nonpredictive actions at each timescale. *** $p < .001$; * $p < .05$; $\sim p < .1$. Source data are provided as a Source Data file.

I wonder to what degree this also holds for the actual sequences used in the experiment. One way to address this concern would be to perform a correlation analysis on the visual stimuli/sequences themselves to determine the amount of overlap in visual stimulation, both when they do and don't share a cue.

We thank the reviewer for this thoughtful suggestion. Certainly, we agree that equating the amount of overlap in visual stimulation across conditions is critical in order to interpret pattern similarity differences in early visual cortex as due to anything other than differences in visual overlap. For this reason, we were very careful to counterbalance the assignment of the visual stimuli across conditions for this study (Methods, page 26).

Notably, while overlap of visual stimulation was purposely equated across conditions for this study, we are currently working on a separate study in which we manipulate the visual similarity of cues and outcomes across different contexts. Indeed, the reviewer's suggestion has directly inspired new and important analyses for that study.

Methods: Stimuli, p. 26, line 11:

“We counterbalanced the assignment of images to 3-day delay and no delay conditions and to sequences containing either predictive or nonpredictive actions, and randomly assigned images to serve as cues or outcomes. The Psychophysics Toolbox⁴² for MATLAB (MathWorks) was used for stimulus presentation and response collection.”

- A direct link between neural and behavioral effects?

It would strengthen the results if the neural effects predicted behavior. Does the strength of hippocampal-neocortical interactions predict behavioral (reaction time) effects, across runs and/or participants?

We thank the reviewer for prompting us to more thoroughly consider possible relationships between behavioral and neural effects. Differences across conditions in background connectivity were not correlated with reaction time in the scanner either across participants or across fMRI runs (Figure S1). Critically, however, a different behavioral measure—consistency of verbal predictions outside of the scanner for nonpredictive actions—is in fact related to background connectivity (Figure 5). This direct link between neural and behavioral effects especially strengthens the results because it suggests a specific mechanism underlying the interaction in background connectivity between timescale and predictiveness. Specifically, background connectivity may have been enhanced above baseline for both conditions immediately after training, while reduced specifically for nonpredictive actions after the 3-day delay. Such changes in consistency could not be examined for predictive actions, because participants were required to be 100% accurate in that condition. We have added a new section to the Results (“Verbal predictions for nonpredictive actions” on page 13) and have expanded

the Discussion (on page 21) in order to incorporate this important interaction between neural and behavioral effects.

Results: Verbal predictions for nonpredictive actions, p. 13, line 14:

“There are multiple potential explanations for the observed interaction between timescale and predictiveness in background connectivity. First, it could be that background connectivity between hippocampus and EVC was at equivalent baseline levels for both predictive and nonpredictive immediately after training, while enhanced specifically for predictive actions after the 3-day delay. Alternatively, it could be that background connectivity was already enhanced above baseline for both predictive and nonpredictive actions immediately after training, while reduced specifically for nonpredictive actions after the 3-day delay. Beyond the control correlations across matched runs that could be used to infer a baseline correlation for the context, behavior on the verbal tests before and after the fMRI scan can be used to help disentangle these possibilities. While participants were required to be 100% accurate in identifying outcomes of predictive actions, there were no correct or incorrect responses for nonpredictive actions. Nonetheless, participants could be consistent or inconsistent in their verbal predictions of unpredictable outcomes. We quantified this behavior for nonpredictive actions based on how consistently each participant mapped each outcome onto specific cue-action combinations (Fig. 5A). In fact, participants were significantly less consistent in verbally identifying expected outcomes of nonpredictive actions learned before the 3-day delay than for nonpredictive actions immediately before the scan ($t(23) = 3.86, p < .001$), suggesting that action-based prediction may have diminished over time for nonpredictive events (Fig. 5B).

Are consistent vs. inconsistent predictions sufficient to modulate hippocampal-neocortical interactions for nonpredictive actions? In total, 14 of the 24 participants were 100% consistent in identifying outcomes for nonpredictive cues and actions immediately after training, while 4 participants were 100% consistent after the 3-day delay. We reasoned that participants who were consistent in verbally identifying the outcomes of nonpredictive actions may have likewise maintained stronger visual predictions for nonpredictive actions than participants who were inconsistent in their responses. If so, such differences across participants may also be reflected in their hippocampal-neocortical interactions. Indeed, background connectivity during nonpredictive actions tended to be greater among participants who made 100% consistent test responses than among participants who made inconsistent responses (Fig. 5C). While this difference between participants was not significant immediately after training ($t(22) = 1.25, p = .22$), it was significant after the 3-day delay ($t(22) = 2.85, p = .009$). Moreover, among participants with 100% consistent test responses, background connectivity was the same for predictive and nonpredictive actions at each timescale ($ps > .79$).”

Figure 5. Verbal predictions for nonpredictive actions. (A) Verbal tests administered immediately before and after each fMRI scan included sequences with nonpredictive actions as well as sequences with predictive actions. (B) Participants more consistently identified particular outcomes for nonpredictive cues and actions learned immediately beforehand than before the 3-day delay. However, a subset of participants for each timescale were 100% consistent in identifying outcomes of nonpredictive actions. (C) Participants who consistently mapped nonpredictive cues and actions to particular outcomes showed greater background connectivity for these nonpredictive events after the 3-day delay. Error bars indicate ± 1 SEM. *** $p < .001$; ** $p < .01$. Source data are provided as a Source Data file.

Discussion: p. 21, line 19:

“Verbal predictions for nonpredictive actions before and after each fMRI scan—and their relationship across participants to background connectivity—support the idea that the hippocampus may at first generate spurious predictions for nonpredictive events. While participants were required to be 100% accurate in identifying outcomes of predictive actions, there were no objectively correct or incorrect responses for nonpredictive actions. However, participants were more than 90% consistent on average in matching nonpredictive cues and actions to specific unpredictable outcomes immediately after training and were significantly less consistent in making such predictions for nonpredictive actions after the 3-day delay. Moreover, the small subset of participants who were still 100% consistent in their verbal predictions for nonpredictive actions after the 3-day delay exhibited significantly stronger background connectivity during nonpredictive events than did participants who made inconsistent predictions.”

- Is enhanced connectivity between hippocampus and visual cortex specific to the retinotopic location of the stimulus?

It would be interesting to see the degree of overlap between voxels that respond to the retinotopic location of the stimuli (as identified by the visual localizer stimulus) and voxels that indicate enhanced connectivity with hippocampus (i.e. visual voxels identified in the partial volume analyses). One way to address this would be to visualize the degree of overlap on the inflated cortical surface.

We thank the reviewer for this suggestion. The revised manuscript includes a new section in the Results (“Specificity within V1/V2” on page 10) in which we address the retinotopic specificity of the background connectivity effects. Since we do not have retinotopic mapping data, we rely on measuring whether voxels showed reliable visual evoked responses to the experimental stimuli in the functional localizer. First, we note that the EVC ROI for all other analyses included all (1.5 mm isotropic) voxels reliably responsive to experimental stimuli in the localizer scan—ranging from 732 to 4,200 voxels or 6.2% to 30.0% of V1/V2. As displayed in Figure 3, we incrementally dilated the size of the EVC ROI from including only the 50 most responsive voxels to including all of V1/V2. The interaction between predictiveness and timescale was significant or marginally significant for ROIs ranging from 50 to 5,000 voxels but not significant for the mean background timeseries of all V1 and V2 voxels. Thus, while differences in hippocampal background connectivity were relatively widespread within V1/V2, they did not extend to all of V1/V2. In turn, the voxelwise background connectivity analyses (Figure 4) reveal the relative specificity of differences in background connectivity with respect to the full field of view of the scan sequence. New results are displayed in Figure 3.

Results: Specificity within V1/V2, p. 10, line 13:

“Are differences in background connectivity specific to only the voxels that are most responsive to the specific retinotopic location of the experimental stimuli, or are they widespread throughout V1/V2? The EVC ROI for each participant included (1.5 mm isotropic) voxels responsive to square- and diamond-masked stimuli in a localizer scan, ranging from 732 to 4,200 voxels in volume (6.2% to 30.0% of V1/V2). To examine the specificity of hippocampal background connectivity within EVC, we varied the extent of the EVC ROI from including just the 50 voxels (<1% of V1/V2) most responsive to functional localizer stimuli to including all V1/V2 voxels (Fig. 3A). Immediately after training, hippocampal background connectivity was equivalent for predictive and nonpredictive actions regardless of the size of the EVC ROI (Fig. 3B). In contrast, after the 3-day delay, background connectivity was reliably stronger for predictive than nonpredictive actions across a wide range of ROI sizes (Fig. 3C). Likewise, the interaction between predictiveness and timescale was significant for ROIs ranging from 50 voxels to 1,000 voxels ($ps < .05$) and marginally reliable for 5,000 voxels ($F(1, 23) = 3.81, p = .06$). However, this interaction was not significant for the mean background timeseries across all V1 and V2 voxels ($F(1, 23) = 0.48, p = .50$). Thus, while differences in hippocampal background connectivity were robust to the size of the EVC ROI, they were not entirely pervasive within V1/V2.”

Figure 3. Specificity within V1/V2. (A) The volume of the EVC ROI was incrementally dilated from including just the 50 voxels most responsive to the experimental stimuli in the functional localizer scan to including all of V1/V2. (B) Across the full range of ROI sizes, background connectivity was equivalent for predictive and nonpredictive actions after no delay. (C) After the 3-day delay, background connectivity was stronger for predictive than nonpredictive actions within ROIs that ranged in volume from 50 voxels to 5,000 voxels but was not reliably different when all V1/V2 voxels were included in the EVC ROI. Error bars indicate ± 1 SEM of the difference between predictive and nonpredictive actions at each timescale. ** $p < .01$; * $p < .05$. Source data are provided as a Source Data file.

Methods: Early visual cortex, p. 32, line 5:

“Additionally, in order to examine the specificity of background connectivity within V1/V2, we incrementally varied the size of the EVC ROI from including all of V1/V2 (mean = 11,828 voxels, s.d. = 2,464 voxels) to including only the 50 voxels in V1/V2 most responsive in an overall contrast of square- and diamond-masked stimuli in the localizer compared to baseline fixation.”

Responses to Reviewer #2

The manuscript by Hindy and colleagues uses human fMRI to probe interactions between hippocampus and early visual cortex during a cue-action-outcome learning task. Prior to fMRI scanning, subjects completed two training sessions (one just before scanning, one 3 days prior to scanning) during which they repeatedly saw various fractals and had to choose an action (left or right response) that led to an outcome. For half of the fractals, responses were highly predictive of outcomes (95% predictive) whereas for the other half of fractals, responses were non-predictive of outcomes (50% predictive). Thus, the two critical factors were whether a given cue was a predictive vs. non-predictive cue and then whether the cue was learned 3 days prior or just prior (same day) to scanning. During scanning, subjects essentially completed the same task with alternating blocks of predictive vs. non-predictive cues, and 3-day vs. same-day cues.

Behaviorally, subjects' reaction times in selecting actions did not differ for predictive vs. non-predictive cues learned the same day as scanning. For cues from the 3-day condition, however, subjects were markedly faster to make decisions in the predictive condition compared to the non-predictive condition. To be clear, this does not mean they were ‘better’—simply that they spent less time deliberating about which response to make and, by extension, which outcome to ‘reveal.’ The primary fMRI measure that is used is background connectivity, which is a measure of the correlation between BOLD responses in different ROIs over time, after regressing out

task-related activity. The idea is that this background connectivity is a measure of communication between regions that is not explained by task-evoked activity. Interestingly, background connectivity between the hippocampus and early visual cortex showed an interaction between delay and predictiveness. For predictive cues, connectivity tended to be stronger for the 3-day cues whereas for non-predictive cues, connectivity tended to be weaker for the 3-day cues. In other words, a longer delay seemed to increase connectivity for predictive cues but weaken connectivity for non-predictive cues. Another potentially interesting finding was that pattern similarity between shared cues with different outcomes tended to decrease in the 3 day condition (compared to the same day condition) but only for predictive cues. This was somewhat true in hippocampus, and in EVC. Interestingly, however, at least for the hippocampus, pattern similarity among non-shared cues showed a qualitatively different pattern with no evidence of a decrease in similarity for the 3 day condition.

While the manuscript addresses an interesting topic (delay-dependent changes in hippocampal-cortical interactions), I have several concerns that limit my enthusiasm. First, the results are discussed in terms of hippocampal predictions, but there is remarkably little in the results/analyses to compel or motivate a “prediction” account. I realize this interpretation is related to a previous, related paper, but the current study just doesn’t seem to have any clear evidence that predictions are even occurring. Second, for several of the analyses, I have questions and/or concerns about how the analyses were performed and whether the interpretations are appropriate.

We thank the reviewer for the highly constructive feedback that has helped us significantly improve the manuscript. We thoroughly address all specific concerns and questions in our responses below. Furthermore, we now strengthen the direct evidence of prediction by relating background connectivity to performance on verbal outcome prediction tests outside of the scanner. This behavioral data is described in a new section of the Results called “Verbal predictions for nonpredictive actions” that was inspired by comment 5 from this reviewer. Please find details about these new results in our response to comment 5 below.

Comments:

1. The idea of using background connectivity to measure “predictions” that are generated by the hippocampus and then sent to EVC is not particularly intuitive to me. I can understand why we would expect predictions to be generated during the actual trials, but why would predictions also be going on “in the background?” Put another way, the fact that global BOLD signal in hippocampus is correlated with global BOLD signal in early visual cortex AFTER regressing out the task effects does not seem like an obvious marker of predictions being sent between these regions. Indeed, prior work from these authors has used much more direct measures of prediction (e.g., by measuring reinstatement of predicted activity patterns). No clear rationale is provided as to why background connectivity is the right measure for the current question.

We thank the reviewer for pointing out the missing justification for measuring background connectivity in the first place. This very important point was also raised by Reviewer #3. Based on the feedback from both reviewers, we have revised the Introduction and Discussion to include rationale for why background connectivity is appropriate for measuring hippocampal involvement in predictive coding and how this involvement may change as a function of time.

In the Introduction (on page 3) as well as the Discussion (on page 25), we detail in two ways the rationale and advantage to measuring background connectivity.

First, we hypothesize that the intrinsic coupling of the hippocampus and EVC, beyond correlations in stimulus-evoked information, may be enhanced when action-based prediction is hippocampally mediated. We motivate this hypothesis by findings in human neurophysiology that link perceptual inference to long-range oscillatory synchronization between the hippocampus and visual cortex (Sehatpour et al., 2008, *PNAS*), together with the observation that stimulus-evoked responses and coherent spontaneous fluctuations are linearly superimposed in human fMRI data (Fox et al 2006, *Nature Neuroscience*).

Second, we now include specific rationale for why background connectivity in particular may be useful for probing hippocampal involvement in perception as a function of time, beyond measurements of stimulus-evoked activity. While correlated classification of stimulus-evoked responses is suggestive of hippocampal-neocortical interactions, such correlations depend upon the fidelity and precision of mnemonic representations in both the hippocampus and visual cortex. Because background correlations may more directly reflect hippocampal-neocortical interactions themselves, we reasoned that it may provide a more general index of hippocampal involvement in perception across different memory-retrieval contexts.

Introduction, p. 3, line 20:

“Beyond correlations in stimulus-evoked information, we hypothesized that the intrinsic coupling of the hippocampus and EVC may be enhanced during action-based prediction. This hypothesis is motivated by findings in human neurophysiology that link perceptual inference to long-range oscillatory synchronization between the hippocampus and visual cortex^{6,7}, together with the observation that stimulus-evoked responses and coherent spontaneous fluctuations are linearly superimposed in human fMRI data⁸. Critically, although correlated classification of stimulus-evoked responses is suggestive of hippocampal-neocortical interactions, such correlations depend upon the precision of memories and associated predictions represented within each region. Therefore, along with measuring multivariate patterns in the hippocampus and EVC, here we used a “background connectivity” approach to quantify the temporal dynamics and covariance of these regions after removing stimulus-evoked responses and other confounding variables^{9,10}. Because background connectivity may more directly measure hippocampal-neocortical interactions than stimulus-specific decoding on its own, we reasoned that it would provide an objective index of the contexts in which the hippocampus is and is not involved in action-based predictive coding.”

Discussion: p. 25, line 1:

“Hippocampal-neocortical interactions measured here through background connectivity are consistent with previous findings in human neurophysiology that link perceptual inference to the synchronization of long-range hippocampal-cortical oscillations^{6,7}. Because stimulus-evoked responses and coherent spontaneous fluctuations are linearly superimposed in human fMRI data⁸, intrinsic activity within the hippocampus and EVC can be separated from stimulus-evoked responses and other variables^{9,10}. And, whereas correlations in classification of stimulus-evoked responses depend upon the precision of memories and associated predictions represented within each region, background correlations may more directly reflect hippocampal-neocortical interactions themselves. In this way, background connectivity provides a more objective index of hippocampal involvement in action-based predictive coding. By using background connectivity to reveal consolidation-related effects on visual prediction,

findings here further develop the link between hippocampal representation^{2,39} and models of predictive coding in visual cortex^{40,41}.”

2. The changes in background connectivity strongly parallel the differences in reaction times across conditions—to a degree that raises serious concerns about what the background connectivity is actually reflecting. Specifically, this raises the obvious concern that the background connectivity measure is affected by reaction time. I’m not exactly sure how to test this with the current data, but it seems like a potentially serious issue. To be clear, I think the behavioral data are interesting, but given that the background connectivity approach involves regressing out task effects, and the task effects are based on trials with very substantial reaction time differences, it seems entirely plausible that the residuals would be influenced by the length of the trial that is being regressed out. Thus, it is hard to imagine that these RT differences would NOT influence the connectivity measures. If the RT differences do drive the differences in background connectivity, then it is obviously not appropriate to interpret the background connectivity as a measure of ‘communication’ between regions; rather it could just be a statistical artifact of regressing out task-related activity when the task conditions differ in reaction times.

We thank the reviewer for raising the important consideration that RT differences in the scanner could directly affect background connectivity and other neural measures. We have addressed this concern in two ways:

First, in Figure S1, we show that background connectivity and RT are not directly related either across participants or across runs. That is, RT variability across participants did not reliably predict the strength of the background connectivity effect (Predictive - Nonpredictive) in either the no-delay condition or after the 3-day delay. Likewise, RT variability across fMRI runs for each participant also did not background connectivity in either delay condition in either the no-delay condition or after the 3-day delay.

Second, in Figure S3, we reveal that background connectivity is however related across participants to pattern similarity in the hippocampus.

Results: ROI background connectivity, p. 8, line 24:

“Furthermore, although the interactions between timescale and predictability in background connectivity paralleled interactions in RT, differences among conditions in background connectivity were not correlated with RT either across participants or across runs for each participant ($ps > .27$; Fig. S1).”

Figure S1. RT vs. background connectivity. (A) Individual differences across participants in RT (Predictive – Nonpredictive) were unrelated to differences in background connectivity immediately after training ($r(22) = -.21, p = .33$) and after the 3-day delay ($r(22) = -.02, p = .94$). (B) In order to examine background connectivity across fMRI runs as a function of the RT, we ranked the runs for each participant based on the RT difference between predictive and nonpredictive actions before plotting background connectivity. For each delay condition, fMRI runs are ranked such that the RT was usually slower for predictive vs. nonpredictive actions for the leftmost columns, and usually faster for predictive than nonpredictive actions for the rightmost columns. In repeated measures ANOVAs based on the ranked fMRI runs, predictiveness did not interact with the RT-rank either immediately after training ($F(1, 23) = 0.38, p = .55$) or after the 3-day delay ($F(1, 23) = 1.29, p = .27$). Error bars indicate ± 1 SEM of the difference between predictive and nonpredictive actions for each run. $*p < .05$; $\sim p < .1$. Source data are provided as a Source Data file.

Results: Multivariate pattern similarity, p. 19, line 14:

“Finally, multivariate pattern similarity in the hippocampus was correlated across participants with background connectivity only after the 3-day delay. Individual differences across participants in background connectivity were unrelated immediately after training to within-cue pattern similarity in either the hippocampus ($r(22) = .11, p = .62$) or EVC ($r(22) = .10, p = .63$; Fig. S3A). In contrast, after the 3-day delay, background connectivity was significantly negatively correlated with pattern similarity in the hippocampus ($r(22) = -.62, p = .001$) though not EVC ($r(22) = -.21, p = .32$; Fig S3B).”

Figure S3. Background connectivity vs. within-cue similarity. (A) Immediately after training, differences in background connectivity between predictive and nonpredictive actions were not reliably correlated across participants with the within-cue pattern similarity of either the hippocampus ($r(22) = .11, p = .62$) or EVC ($r(22) = .10, p = .63$). (B) After the 3-day delay, the differences in background connectivity between predictive and nonpredictive actions were negatively correlated across participants with the within-cue pattern similarity of the hippocampus ($r(22) = -.62, p = .001$) but not EVC ($r(22) = -.21, p = .32$). $^{**}p < .01$. Source data are provided as a Source Data file.

3. Was Distinct cue similarity always based on cues that had different button presses (i.e., left vs. right). If not, then the relatively greater similarity among distinct cues could obviously be an artifact of the fact that the Different cue similarity includes trials with a common motor response whereas the Shared cue trials were necessarily restricted to trials that had opposite motor responses.

We thank the reviewer for drawing our attention to missing details in the original description of the multivariate similarity analyses. In the revised Methods, on page 35, we clarify that within-cue and across-cue similarity were calculated in exactly the same way. Thus, for predictive actions, across-cue pattern similarity was measured across left and right button presses in the

same way as within-cue pattern similarity. For nonpredictive actions, we averaged across left and right button presses for the within-cue and across-cue similarity. Importantly, “predictive vs. nonpredictive” actions significantly interacted with “within-cue vs. across-cue” pattern similarity in both the hippocampus ($p = .007$) and EVC ($p = .003$).

In addition to reliable interactions between predictiveness and within-cue/across-cue similarity, follow-up analyses further ensured that averaging across left and right button presses for nonpredictive events did not bias contrasts between predictive and nonpredictive actions (described on page 35 of the Methods and displayed in Figure S2: “Resampled pattern similarity and contrast-to-noise ratios”; please see above in response to Reviewer #1). For these follow-up analyses we calculated pattern similarity specifically between cue-action-outcome sequences that differed in both the action and the outcome. After equating voxel-level variability for predictive and nonpredictive actions (Figure S2B), all of the original within-cue pattern similarity effects were also found in these follow-up analyses.

Results: Multivariate pattern similarity, p. 17, line 13:

“Critically, visual stimulation was the same for cue-outcome transitions containing either predictive or non-predictive actions: either of two outcomes followed a cue and double-sided arrow with equal probability. But since nonpredictive actions could not be decoded in either the hippocampus or EVC in a prior study with the same action-based prediction task³, we averaged across left and right button presses for visual transitions that contained nonpredictive actions. However, all of the pattern similarity effects replicated in follow-up analyses with resampled data that split between left and right nonpredictive actions (Fig. S2). Moreover, predictiveness significantly interacted in each ROI with comparisons of within-cue vs. across-cue pattern similarity of the exact same multivoxel patterns ($ps < .01$; Fig. 7B).”

Results: Multivariate pattern similarity, p. 18, line 21:

“However, as noted above, within-cue vs. across-cue similarity reliably interacted with predictiveness in both the hippocampus ($F(1, 23) = 8.95, p = .007$) and EVC ($F(1, 23) = 11.33, p = .003$).”

Methods: Multivariate pattern similarity, p. 35, line 4:

“Within-cue similarity was measured as the correlation between cue-outcome transitions containing the same cue but different outcomes (e.g., “A1-left-B1 vs. A1-right-C1” for predictive actions and “D1-[left/right]-E1 vs. D1-[left/right]-F1” for nonpredictive actions). Across-cue pattern similarity was measured as the correlation between cue-outcome transitions containing completely distinct cue and outcome stimuli (e.g., “A1-left-B1 vs. A2-right-B2” for predictive actions and “D1-[left/right]-E1 vs. D2-[left/right]-F2” for nonpredictive actions). For cue-outcome transitions with predictive actions, across-cue pattern similarity was measured across left and right button presses in the same way as within-cue pattern similarity.”

Methods: Multivariate pattern similarity, p. 35, line 24:

“As a follow-up control analysis to ensure that averaging across left and right button presses for visual transitions with nonpredictive actions did not bias the primary findings, we additionally calculated the within-cue pattern similarity between visual transitions with left vs. right button presses for nonpredictive actions (Fig. S2). For this analysis, each nonpredictive cue-action-outcome sequence was modeled with its own regressor and temporal derivative (e.g., D1-left-E1, D1-left-F1, D1-right-E1, and D1-right-F1). Within-cue similarity was then measured specifically between cue-action-outcome sequences that shared the same cue but differed in

both the action and the outcome (e.g., “D1-left-E1 vs. D1-right-F1” and “D1-left-F1 vs. D1-right-E1”). Notably, splitting up trials in this way for nonpredictive actions cut in half the number of trials for estimating the neural pattern for each cue-outcome visual transition (from about 5 trials per run in the primary analyses to about 2.5 trials per run in the resampled analysis). In order to calculate pattern similarity in the same way for visual transitions containing either predictive or nonpredictive actions (and thereby equate CNR across conditions), we randomly resampled trials with predictive actions as belonging to either the “Rand a” or “Rand b” dataset. Then, we calculated within-cue pattern similarity separately for each randomly sampled dataset (e.g., “A1a-left-B1a vs. A1a-right-C1a” and “A1b-left-B1b vs. A1b-right-C1b”) before averaging together correlation coefficients in the same way as for visual transitions containing nonpredictive actions.”

4. The pattern similarity results are interesting, but not overwhelmingly compelling. For one, the hippocampal interaction for shared cue data is only marginally significant. Additionally, for EVC, although there is a significant interaction between delay and predictiveness for the within-cue data and not for the across-cue data, it seems extremely unlikely that the 3-way interaction between delay, predictiveness and within-cue/across-cue is significant. For hippocampus, this three-way interaction might well be significant, though I don’t think it is reported. But EVC certainly seems to show a very similar pattern for the within-cue and the across-cue analyses.

We thank the reviewer for encouraging us to think more critically about statistical interactions in the pattern similarity results. In the revised Results, on pages 18 and 19, we explicitly acknowledge that the hippocampal interaction was marginally significant ($p = .051$), and we are more thorough in our reporting of relevant statistical interactions. Specifically, we now additionally report statistics based on 3-factor ANOVAs of predictiveness, timescale, and within-cue vs. across-cue similarity. Although 3-way interactions were not reliable in either ROI ($ps > .12$), these ANOVAs revealed that predictiveness reliably interacted with comparisons of within-cue vs. across-cue similarity in both the hippocampus ($p = .007$) and EVC ($p = .003$). Based on the overall body of evidence across conditions and ROIs, we think the pattern similarity results are at least helpful for interpreting the primary findings related to hippocampal-neocortical background connectivity.

Results: Multivariate pattern similarity, p. 18, line 2:

“Importantly, differentiation effects were modulated by delay condition, including a marginally reliable interaction between predictiveness and timescale in the hippocampus ($F(1, 23) = 4.26$, $p = .051$) and a significant interaction in EVC ($F(1, 23) = 5.36$, $p = .03$).”

Results: Multivariate pattern similarity, p. 18, line 21:

“However, as noted above, within-cue vs. across-cue similarity reliably interacted with predictiveness in both the hippocampus ($F(1, 23) = 8.95$, $p = .007$) and EVC ($F(1, 23) = 11.33$, $p = .003$). Unlike the differentiation effect between overlapping cue-outcome transitions, across-cue similarity did not interact with delay condition in either the hippocampus ($F(1, 23) = 1.74$, $p = .20$) or EVC ($F(1, 23) = 0.96$, $p = .34$), though the 3-way interaction of within-cue vs. across-cue similarity, predictiveness, and timescale was not reliable in either ROI ($ps > .12$).”

5. I was not fully clear on how the pattern similarity analyses were performed. Was there a separate parameter estimate for each trial in each run? Were correlations ever obtained between

data from the same run, or was this restricted to across-run correlations? If same-run analyses were performed, are there any differences between the shared-cue and across-cue analyses in terms of lag effects? For example, depending on whether or not the trial order included consecutive repeats of the same condition, there might be a bias such that across-cue analyses were systematically based on trials that were further apart (in terms of mean trial lag), which could artificially make the across-cue similarity values increase. In any case, more detail is required about these analyses.

We thank the reviewer for prompting us to clarify how the multivariate pattern analyses were performed and acknowledge that the description in the original manuscript was not adequately detailed. In short, each cue-outcome transition (e.g., “A1-left-B1” or “D1-[left/right]-E1”) was modeled within each run as a single regressor (along with its temporal derivative) so that a single parameter estimate was calculated at each voxel for each visual transition. This resulted in 8 patterns per run: 4 patterns for the cue-outcome transitions associated with predictive actions and 4 patterns for the cue-outcome transitions associated with nonpredictive actions. Within-cue and across-cue correlations reflecting pattern similarity were calculated after averaging these patterns across runs for each of the 8 cue-outcome transitions. Importantly, the trial order of the cue stimuli was randomized within and across blocks of both predictive and nonpredictive actions within each run. This randomization ensured that there were no systematic differences in trial lag for the within-cue (e.g., “A1-left-B1” vs. “A1-right-B1”) and across-cue (e.g., “A1-left-B1” vs. “A2-right-B2”) comparisons. We have expanded the Methods (on pages 34 and 35) in order to explain in detail exactly how the multivariate patterns of activity were estimated and compared, and therein specifically address each of the reviewer’s questions along with other details about the analysis.

Methods: Scan task, p. 28, line 15:

“Pairs of runs for each participant contained the same stimuli and block order, while the trial order of the cue stimuli was randomized within and across blocks of predictive or nonpredictive actions. For nonpredictive actions, the trial order of the associated outcomes was also randomized within and across blocks.”

Methods: Multivariate pattern similarity, p. 34, line 13:

“Multivoxel patterns in the hippocampus and EVC for each cue-outcome visual transition were based on parameter estimates of BOLD response amplitude in an event-related GLM for each run. Each cue-outcome transition was modeled with its own regressor and temporal derivative, constructed by convolving a boxcar function that matched the average trial duration for the condition (between 2188 and 2643 ms, depending on the participant’s mean response time) with a double-gamma hemodynamic response function. This resulted in 8 regressors of interest: 4 regressors for the cue-outcome transitions associated with predictive actions (e.g., A1-left-B1 and A1-right-C1), and 4 regressors for the cue-outcome transitions associated with nonpredictive actions (e.g., D1-[left/right]-E1 and D1-[left/right]-F1). Each GLM was fit using FILM with local autocorrelation correction and six motion parameters as nuisance covariates, as well as an additional regressor and its temporal derivative to model the single predictive event within each run that contained a counter-predicted outcome, along with trials for which the participant failed to press a button before the response deadline. Parameter estimates for each visual transition were then averaged across runs before calculating pattern similarity.

Pattern similarity was measured as z -transformed Pearson correlations across voxels within each ROI. Within-cue similarity was measured as the correlation between cue-outcome transitions containing the same cue but different outcomes (e.g., “A1-left-B1 vs. A1-right-C1”

for predictive actions and “D1-[left/right]-E1 vs. D1-[left/right]-F1” for nonpredictive actions). Across-cue pattern similarity was measured as the correlation between cue-outcome transitions containing completely distinct cue and outcome stimuli (e.g., “A1-left-B1 vs. A2-right-B2” for predictive actions and “D1-[left/right]-E1 vs. D2-[left/right]-F2” for nonpredictive actions). For cue-outcome transitions with predictive actions, across-cue pattern similarity was measured across left and right button presses in the same way as within-cue pattern similarity.”

6. There is an interesting idea raised in the Discussion that in the no delay condition, the hippocampus might still be generating predictions, but they are as likely to be correct as incorrect which is why there is no difference for the predictive vs. non-predictive conditions. Yet, this argument seems to be contradicted by the behavioral data in that subjects were, in fact, required to reach 100% accuracy in a test of prediction memory before entering the scanner. Obviously, there is a change in behavior over time (reflected in RTs) and I do think that change is interesting, but given that accuracy was forced to be at ceiling, that seems to argue against the idea that subjects are generating inaccurate predictions in the no delay condition.

We are grateful to the reviewer for raising this concern and orienting our attention to the verbal tests outside the scanner as providing a key behavioral measure of prediction for nonpredictive actions (not just predictive actions). Indeed, participants were required to be 100% accurate in identifying outcomes of predictive actions. However, accuracy could not be objectively measured for nonpredictive actions because there was no correct or incorrect response for each trial.

However, even though there were no objectively correct or incorrect responses for nonpredictive actions in these tests, participants could nonetheless be consistent or inconsistent in their verbal predictions for each cue and action. We quantified how consistently each participant mapped each outcome onto specific cue-action combinations. Participants were significantly less consistent in verbally identifying expected outcomes of nonpredictive actions learned before the 3-day delay than for nonpredictive actions immediately before the scan (Figure 5: “Verbal predictions for nonpredictive actions”; please see above in response to Reviewer #1). Importantly, this result is consistent with the idea that action-based prediction may have diminished over time for nonpredictive events.

Furthermore, behavioral differences in verbal prediction were related across participants to differences in background connectivity for nonpredictive actions. Those participants who were 100% consistent in their verbal predictions for nonpredictive actions also displayed significantly stronger background connectivity during these events in the scanner than did participants who made inconsistent predictions, especially after the 3-day delay. In the revised manuscript, we have added a new section to the Results (“Verbal predictions for nonpredictive actions” on page 13) and have expanded the Discussion (on page 21) in order to incorporate this important interaction between neural and behavioral effects.

Results: Verbal predictions for nonpredictive actions, p. 13, line 14:

“There are multiple potential explanations for the observed interaction between timescale and predictiveness in background connectivity. First, it could be that background connectivity between hippocampus and EVC was at equivalent baseline levels for both predictive and nonpredictive immediately after training, while enhanced specifically for predictive actions after the 3-day delay. Alternatively, it could be that background connectivity was already

enhanced above baseline for both predictive and nonpredictive actions immediately after training, while reduced specifically for nonpredictive actions after the 3-day delay. Beyond the control correlations across matched runs that could be used to infer a baseline correlation for the context, behavior on the verbal tests before and after the fMRI scan can be used to help disentangle these possibilities. While participants were required to be 100% accurate in identifying outcomes of predictive actions, there were no correct or incorrect responses for nonpredictive actions. Nonetheless, participants could be consistent or inconsistent in their verbal predictions of unpredictable outcomes. We quantified this behavior for nonpredictive actions based on how consistently each participant mapped each outcome onto specific cue-action combinations (Fig. 5A). In fact, participants were significantly less consistent in verbally identifying expected outcomes of nonpredictive actions learned before the 3-day delay than for nonpredictive actions immediately before the scan ($t(23) = 3.86, p < .001$), suggesting that action-based prediction may have diminished over time for nonpredictive events (Fig. 5B).

Are consistent vs. inconsistent predictions sufficient to modulate hippocampal-neocortical interactions for nonpredictive actions? In total, 14 of the 24 participants were 100% consistent in identifying outcomes for nonpredictive cues and actions immediately after training, while 4 participants were 100% consistent after the 3-day delay. We reasoned that participants who were consistent in verbally identifying the outcomes of nonpredictive actions may have likewise maintained stronger visual predictions for nonpredictive actions than participants who were inconsistent in their responses. If so, such differences across participants may also be reflected in their hippocampal-neocortical interactions. Indeed, background connectivity during nonpredictive actions tended to be greater among participants who made 100% consistent test responses than among participants who made inconsistent responses (Fig. 5C). While this difference between participants was not significant immediately after training ($t(22) = 1.25, p = .22$), it was significant after the 3-day delay ($t(22) = 2.85, p = .009$). Moreover, among participants with 100% consistent test responses, background connectivity was the same for predictive and nonpredictive actions at each timescale ($ps > .79$).”

Discussion: p. 21, line 19:

“Verbal predictions for nonpredictive actions before and after each fMRI scan—and their relationship across participants to background connectivity—support the idea that the hippocampus may at first generate spurious predictions for nonpredictive events. While participants were required to be 100% accurate in identifying outcomes of predictive actions, there were no objectively correct or incorrect responses for nonpredictive actions. However, participants were more than 90% consistent on average in matching nonpredictive cues and actions to specific unpredictable outcomes immediately after training and were significantly less consistent in making such predictions for nonpredictive actions after the 3-day delay. Moreover, the small subset of participants who were still 100% consistent in their verbal predictions for nonpredictive actions after the 3-day delay exhibited significantly stronger background connectivity during nonpredictive events than did participants who made inconsistent predictions.”

Responses to Reviewer #3

In this manuscript, Hindy et al. present an intriguing follow-up to their prior work on a hippocampal role in predictive coding in visual cortex. They find that predictive cue-action-outcome sequences learned 3 days before fMRI scanning are associated with faster responses,

greater state-based connectivity between hippocampus (HPC) and early visual cortex (EVC), and more dissimilar neural patterns in HPC and EVC versus sequences learned on the same days as scanning. Both the predictiveness of the sequences and the 3-day delay in learning had significant impacts on all measures. The authors argue that these findings point to a specific role for the HPC in binding cue-action-outcomes that exhibit regularity and that interactions between HPC and EVC for these predictive events strengthen over time.

Overall, this work is timely, well-conducted, and novel. The findings coalesce into a consistent package with fairly straightforward theoretical implications. This work will undoubtedly impact the field and motivate new research into the HPC as one of the brain's engines for prediction. There are, however, several issues detailed below that should be fully considered.

We are delighted by the reviewer's enthusiasm and positive assessment of the manuscript and are very appreciative of their thoughtful comments and advice. Indeed, we have followed the reviewer's advice on how to reframe the paper to focus on its novel contribution, how to more clearly motivate the background connectivity approach, how to clarify sections of the analysis, and how to be more explicit about the broader theoretical implications of the findings.

Major concerns

1) After reading the introduction, I already had the overall gist of my review in mind: too incremental. However, after reading the manuscript in its entirety, it is clearly much more than a simple follow-up to the authors' prior's work and represents a novel contribution in its own right. As such, I think the introduction does not appropriately set the stage for the research. Although the authors have attempted to answer big important questions, the introduction has a much too narrow focus on their Nature Neuroscience paper. As is, the intro is not necessarily wrong, but it fails to motivate the larger question of hippocampal-cortical interactions for predictive coding to the more general audience of Nature Communications readers.

We thank the reviewer for their valuable suggestion to reframe the Introduction for the broad audience of *Nature Communications* and are pleased that they recognized the novelty and significance of the study despite the overly conservative framing of the original submission. Taking the reviewer's advice to heart, we have substantially revised the Introduction to more directly focus on the novel contribution and larger significance of the findings. We still include an abbreviated description of the previous findings based on the multivariate decoding of stimulus representations. However, we use these findings to motivate our more general focus intrinsic coupling and background connectivity as key to deciphering how hippocampal-neocortical interactions may provide a neural mechanism for predictive coding (as opposed to using these findings to motivate a specific follow-up to the previous study). From there, we more appropriately focus on the timescale manipulation as providing a general test between alternative models of the hippocampal function (as opposed to testing between alternative interpretations of the prior findings). Again, we thank the reviewer for their valuable advice and appreciate the opportunity to revise our manuscript accordingly.

2) The authors motivate the study with two competing accounts of HPC's involvement in predictive coding: 1) HPC is endowed with the computations for prediction, thus should always be engaged in learning regularities and 2) HPC's role in prediction is restricted to early learning before consolidation at which point the baton is passed to cortical processes. Reasonably, they

then run a task with manipulations of time thereby allowing for a test of two accounts. However, their findings do not seem to align well with either of the accounts. HPC processes and representations are not involved immediately after learning (thus ruling out account 1), but are involved after a 3 day delay (providing support against account 2). Although these findings are consistent with the authors' non-specific predictions that HPC-EVC interactions would depend on lag and predictiveness, they do not support either of the competing accounts. What updates in theory are needed to support the current findings? The authors provide hints of this in the discussion, but an explicit appreciation of this divergence from more conventional views of HPC function should be expanded.

We thank the reviewer for these important points about the theoretical framing of the paper. We recognize that, although the Introduction had framed the experiment as a test between two competing accounts of hippocampal involvement in predictive coding, these two alternative accounts were never explicitly revisited. Indeed, neither of the alternative accounts would obviously predict identical hippocampal-neocortical interaction for predictive and nonpredictive actions immediately after training. We now directly address divergence of the findings from more conventional account of hippocampal functions, including a discussion of additional analyses and results that suggest specific updates in theory in order to support the current findings.

In a new section of the Results beginning on page 13 and including the new Figure 5 (“Verbal predictions for nonpredictive actions”; please see above in response to Reviewer #1), we reveal how a relationship between hippocampal-neocortical interactions and verbal predictions for nonpredictive actions may help reconcile findings with models that suggest a sustained role for the hippocampus in memory retrieval. Specifically, new findings support the idea that the hippocampus may at first generate predictions even for nonpredictive events.

In the revised Discussion, on page 23, we explicitly address how neural and behavioral findings diverge from conventional views of hippocampal function, and we outline updates to these theories in order to reconcile them with the current data. Ideas that were hinted at in the original submission are now more appropriately framed and contextualized.

Discussion: p. 23, line 7:

“At the same time that changes across time in hippocampal-neocortical interaction are inconsistent with a time-invariant role for the hippocampus in predictive coding, models of memory retrieval that posit a reduced role for the hippocampus over time¹¹⁻¹³ would not obviously predict the findings: identical hippocampal-neocortical interaction during predictive and nonpredictive actions immediately after training followed by greater interaction specifically during predictive actions after a 3-day delay. In order to accommodate these findings, models that include the hippocampus need to include a role for predictive action in offline processing. Specifically, predictive action may provide a mechanism for prioritizing which representations are either strengthened through synaptic potentiation or weakened through synaptic depression during periods of offline rest^{29,30}. Activity-dependent synaptic potentiation and depression may in turn be mediated by offline replay within the hippocampus^{31,32} and between the hippocampus and neocortex^{33,34}. By transforming noisy recent associations into sparser remote associations, this offline processing may increase the efficiency and utility of hippocampal associations over time^{35,36}. Ultimately, sparser hippocampal representations may increase the signal-to-noise of the hippocampal-neocortical interactions during action-based prediction.”

Results: Verbal predictions for nonpredictive actions, p. 13, line 14:

“There are multiple potential explanations for the observed interaction between timescale and predictiveness in background connectivity. First, it could be that background connectivity between hippocampus and EVC was at equivalent baseline levels for both predictive and nonpredictive immediately after training, while enhanced specifically for predictive actions after the 3-day delay. Alternatively, it could be that background connectivity was already enhanced above baseline for both predictive and nonpredictive actions immediately after training, while reduced specifically for nonpredictive actions after the 3-day delay. Beyond the control correlations across matched runs that could be used to infer a baseline correlation for the context, behavior on the verbal tests before and after the fMRI scan can be used to help disentangle these possibilities. While participants were required to be 100% accurate in identifying outcomes of predictive actions, there were no correct or incorrect responses for nonpredictive actions. Nonetheless, participants could be consistent or inconsistent in their verbal predictions of unpredictable outcomes. We quantified this behavior for nonpredictive actions based on how consistently each participant mapped each outcome onto specific cue-action combinations (Fig. 5A). In fact, participants were significantly less consistent in verbally identifying expected outcomes of nonpredictive actions learned before the 3-day delay than for nonpredictive actions immediately before the scan ($t(23) = 3.86, p < .001$), suggesting that action-based prediction may have diminished over time for nonpredictive events (Fig. 5B).

Are consistent vs. inconsistent predictions sufficient to modulate hippocampal-neocortical interactions for nonpredictive actions? In total, 14 of the 24 participants were 100% consistent in identifying outcomes for nonpredictive cues and actions immediately after training, while 4 participants were 100% consistent after the 3-day delay. We reasoned that participants who were consistent in verbally identifying the outcomes of nonpredictive actions may have likewise maintained stronger visual predictions for nonpredictive actions than participants who were inconsistent in their responses. If so, such differences across participants may also be reflected in their hippocampal-neocortical interactions. Indeed, background connectivity during nonpredictive actions tended to be greater among participants who made 100% consistent test responses than among participants who made inconsistent responses (Fig. 5C). While this difference between participants was not significant immediately after training ($t(22) = 1.25, p = .22$), it was significant after the 3-day delay ($t(22) = 2.85, p = .009$). Moreover, among participants with 100% consistent test responses, background connectivity was the same for predictive and nonpredictive actions at each timescale ($ps > .79$).”

3) Why background connectivity? Don't worry, I'm a fan of background connectivity and think the approach is underappreciated in the field. But, very little is provided to justify why background connectivity is a good measure for this study. More importantly, what do the connectivity findings imply at a mechanistic level, especially in light of their previous findings? I think the preferred argument is that this coupled activity arises due to HPC-guided cortical reinstatement. A more thorough motivation for using background connectivity and speculation for why functional coupling is mechanistically important for predictive coding would strengthen the manuscript and potentially encourage wider adoption of such connectivity methods.

We thank the reviewer for insight into how to strengthen the justification for focusing specifically on background connectivity (a point that was also raised by Reviewer #2). In accord with the reviewer's suggestions, we have revised the Introduction and Discussion to include focus on the mechanistic implications of background connectivity. Specifically, we

relate background connectivity to evidence from human neurophysiology that long-range oscillatory synchronization between the hippocampus and visual cortex is related to perceptual inference (Sehatpour et al., 2008, *PNAS*). Furthermore, we now include specific rationale for why background connectivity in particular is an appropriate measure for our specific questions about how hippocampal involvement in predictive coding may change as a function of time.

Introduction, p. 3, line 20:

“Beyond correlations in stimulus-evoked information, we hypothesized that the intrinsic coupling of the hippocampus and EVC may be enhanced during action-based prediction. This hypothesis is motivated by findings in human neurophysiology that link perceptual inference to long-range oscillatory synchronization between the hippocampus and visual cortex^{6,7}, together with the observation that stimulus-evoked responses and coherent spontaneous fluctuations are linearly superimposed in human fMRI data⁸. Critically, although correlated classification of stimulus-evoked responses is suggestive of hippocampal-neocortical interactions, such correlations depend upon the precision of memories and associated predictions represented within each region. Therefore, along with measuring multivariate patterns in the hippocampus and EVC, here we used a “background connectivity” approach to quantify the temporal dynamics and covariance of these regions after removing stimulus-evoked responses and other confounding variables^{9,10}. Because background connectivity may more directly measure hippocampal-neocortical interactions than stimulus-specific decoding on its own, we reasoned that it would provide an objective index of the contexts in which the hippocampus is and is not involved in action-based predictive coding.”

Discussion: p. 25, line 1:

“Hippocampal-neocortical interactions measured here through background connectivity are consistent with previous findings in human neurophysiology that link perceptual inference to the synchronization of long-range hippocampal-cortical oscillations^{6,7}. Because stimulus-evoked responses and coherent spontaneous fluctuations are linearly superimposed in human fMRI data⁸, intrinsic activity within the hippocampus and EVC can be separated from stimulus-evoked responses and other variables^{9,10}. And, whereas correlations in classification of stimulus-evoked responses depend upon the precision of memories and associated predictions represented within each region, background correlations may more directly reflect hippocampal-neocortical interactions themselves. In this way, background connectivity provides a more objective index of hippocampal involvement in action-based predictive coding. By using background connectivity to reveal consolidation-related effects on visual prediction, findings here further develop the link between hippocampal representation^{2,39} and models of predictive coding in visual cortex^{40,41}.”

4) At first glance, the difference in choice RT for predictive vs. nonpredictive sequences at a delay provides converging evidence for the authors' claims. However, I hesitate on understanding what is driving this effect. Although there appears to be a bit of speeding for the delay predictive sequences, relative to no delay, the majority of this effect seems to be in slower responses to the delay nonpredictive sequences. Given that there are no “wrong” responses, what is slowing RTs to the nonpredictive sequences? In the discussion, the authors do describe an account of hippocampal-based strengthening of sparse representations for predictive associations, which is line with relatively faster responses to predictive sequences. But, what makes delayed nonpredictive slower relative to no delay? It may be that loosely-bound, weak representations of each nonpredictive cue-action-outcome sequence are all retrieved and

compete for action selection. Also, I'm interested to see if individual differences in this behavioural effect are related to neural measures, either differences in connectivity or representational similarity (e.g., are faster RTs correlated with greater pattern dissimilarity for predictive sequences?). In any case, I do think the RT effect is interesting and may shed light on the mechanisms at play, but would appreciate the authors giving this effect more consideration in the discussion.

We thank the reviewer for these mindful questions, and for the suggestion to more directly relate behavioral and neural data. We have expanded the Discussion (on page 22) to include more consideration of the RT differences. Additionally, we have addressed the reviewer's specific questions through new analyses in the Results.

Specifically, on pages 8 and 19 of the Results and in Figure S1 ("RT vs. background connectivity"; please see above in response to Reviewer #2), we show that neither background connectivity nor pattern similarity were correlated RT. At the same time, however, background connectivity and hippocampal pattern similarity were significantly correlated with each other across participants (Figure S3: "Background connectivity vs. pattern similarity"; please see above in response to Reviewer #2).

Additionally, on page 6 of the Results, we separately compared predictive and nonpredictive events across delay conditions. This revealed that while speeded RT over time for predictive actions was marginally significant, slower RT over time for nonpredictive actions was not significant.

Most notably, along with RT, we now report consistency of verbal predictions as an additional behavioral measure that in fact is related background connectivity (Figure 5: "Verbal predictions for nonpredictive actions"; please see above in response to Reviewer #1).

Discussion: p. 22, line 7:

"Three days after training, participants were also significantly quicker in making predictive actions than in making nonpredictive actions. Although RT in the scanner did not correlate with background connectivity across participants, faster overall responses to predictive vs. nonpredictive cues after the delay are consistent with differences across conditions in background connectivity. Accordingly, changes in across time in hippocampal-neocortical interaction may underlie these changes in the perceptual fluency of cue-action-outcome sequences. At the same time that strengthening of sparse hippocampal representations may lead to faster responses to predictive cues, weak or noisy representations for nonpredictive associations may lead to slower responses to nonpredictive cues. While time-dependent changes in perceptual fluency of tasks such as texture discrimination²² and visual contour integration²³ may be independent of the hippocampus, hippocampal function is necessary for learning arbitrary associations among stimuli²⁴. Notably, however, the statistical learning required for action-based prediction may involve different pathways within the hippocampus than other forms of hippocampally dependent learning^{5,24}."

Results: ROI background connectivity, p. 8, line 24:

"Furthermore, although the interactions between timescale and predictability in background connectivity paralleled interactions in RT, differences among conditions in background connectivity were not correlated with RT either across participants or across runs for each participant ($ps > .27$; Fig. S1)."

Results: Multivariate pattern similarity, p. 19, line 14:

“Finally, multivariate pattern similarity in the hippocampus was correlated across participants with background connectivity only after the 3-day delay. Individual differences across participants in background connectivity were unrelated immediately after training to within-cue pattern similarity in either the hippocampus ($r(22) = .11, p = .62$) or EVC ($r(22) = .10, p = .63$; Fig. S3A). In contrast, after the 3-day delay, background connectivity was significantly negatively correlated with pattern similarity in the hippocampus ($r(22) = -.62, p = .001$) though not EVC ($r(22) = -.21, p = .32$; Fig S3B). Like background connectivity, pattern similarity in each ROI was not correlated with individual differences in RT at either timescale ($ps > .08$).”

Results: Choice RT for predictive and nonpredictive actions, p. 6, line 10:

“When predictive and nonpredictive events were separately compared across delay conditions, speeded RT over time for predictive actions was marginally significant ($t(23) = 1.85, p = .08$), while slower RT over time for nonpredictive actions was not significant ($t(23) = 1.57, p = .13$).”

5) Pattern dissimilarity in EVC immediate suggests separation is occurring without the hippocampus. However, whereas the HPC results for the shared cue sequences are clearly different than the distinct cue sequences, EVC shows the same pattern for both. Two questions about the GLMs for estimating neural patterns in this analysis: 1) Were the action arrows included as part of the pattern-defining regressors (i.e., was the entire trial sequence modelled or jus the cue? 2) Were all combinations of action pairs (left-right, left-left, right-right) included in the nonpredictive similarity comparisons? If the answer to both of these is yes, it may be that nonpredictive similarity is biased higher due to the matched visual stimulus from the action arrow (i.e., left-left and right-right trial pairs). The predictive similarity comparisons were always different actions, thus had less objective visual similarity across the sequence.

We thank the reviewer for prompting us to clarify ambiguities about how the pattern similarity analyses were performed. We address each of the reviewer’s specific questions below. First, we now clarify in the Methods (on page 34) the entire trial sequence (cue, action arrow, outcome) was modeled in pattern-defining regressors for the similarity analyses. Critically, however, the action arrow never impacted the visual similarity of the sequences since it was always the same double-sided arrow and participants chose which button to press for each trial. Second, we now explain in the Methods (on page 35) how exactly we calculated and compared multivariate patterns. For all pattern similarity analyses, comparisons between predictive events and comparisons between nonpredictive events were identical with respect to visual stimuli. Third, additional control analyses are displayed in Figure S2 (“Resampled pattern similarity and contrast-to-noise ratios”) in order to address a related consideration regarding motor responses for predictive vs. nonpredictive actions. (Please see our response to comment 1 of Reviewer #1 for more detail on this last point.)

Methods: Multivariate pattern similarity, p. 34, line 13:

“Multivoxel patterns in the hippocampus and EVC for each cue-outcome visual transition were based on parameter estimates of BOLD response amplitude in an event-related GLM for each run. Each cue-outcome transition was modeled with its own regressor and temporal derivative, constructed by convolving a boxcar function that matched the average trial duration for the condition (between 2188 and 2643 ms, depending on the participant’s mean response time) with a double-gamma hemodynamic response function. This resulted in 8 regressors of interest:

4 regressors for the cue-outcome transitions associated with predictive actions (e.g., A1-left-B1 and A1-right-C1), and 4 regressors for the cue-outcome transitions associated with nonpredictive actions (e.g., D1-[left/right]-E1 and D1-[left/right]-F1). Each GLM was fit using FILM with local autocorrelation correction and six motion parameters as nuisance covariates, as well as an additional regressor and its temporal derivative to model the single predictive event within each run that contained a counter-predicted outcome, along with trials for which the participant failed to press a button before the response deadline. Parameter estimates for each visual transition were then averaged across runs before calculating pattern similarity.”

Methods: Multivariate pattern similarity, p. 35, line 12:

“Cue-outcome transitions were visually identical for predictive and nonpredictive actions – either of two outcomes stimuli followed a cue stimulus and a double-sided arrow with equal probability as one another. However, while button presses differed across alternative visual transitions for each predictive cue, this was not the case for nonpredictive cues (in which either button press could produce either outcome). Since nonpredictive actions could not be decoded in either the hippocampus or EVC during action-based prediction in a previous study with the same task paradigm³, we averaged across left and right button presses in order to estimate the multivoxel patterns for visual transitions with nonpredictive actions. Importantly, averaging in this way balanced the number of observations used to estimate neural patterns for each condition, thereby equating the contrast-to-noise ratio (CNR)⁵⁵ of patterns for each condition (Fig. S2B). Equating CNR across conditions is important for comparing pattern similarity across conditions because voxel-level variability strongly influences multivoxel correlations among patterns^{56,57}.”

Assuming this confound isn’t present or impacting the presented results, an extended discussion of these EVC pattern dissimilarity findings seems warranted. Is a different non-hippocampal top-down signal separating EVC patterns early in learning? Or is this an intrinsic computation of EVC?

We thank the reviewer for pointing out this important finding. Indeed, we have substantially revised the manuscript in order to address hypotheses related to this divergence between background connectivity and multivariate pattern analysis. Notably, the finding that hippocampal-neocortical interactions were the same for predictive and nonpredictive actions immediately after training also appears to conflict with our previous finding based on MVPA (Hindy et al 2016, *Nature Neuroscience*).

In the revised Discussion, on page 21, we build upon the idea that background connectivity and pattern analysis may be differentially sensitive to prediction in our task, and that background connectivity between the hippocampus and EVC may have been enhanced for both predictive and nonpredictive actions immediately after training. Importantly, this suggestion is supported by behavioral evidence of prediction for nonpredictive actions immediately after training (Figure 5: “Verbal predictions for nonpredictive actions”; please see above in response to Reviewer #1), along with preliminary evidence that these spurious predictions are in fact related to background connectivity between the hippocampus and EVC.

Discussion: p. 21, line 1:

“Immediately after training, hippocampal-neocortical interactions were the same for predictive and nonpredictive actions. At first glance, the absence of a difference in background

connectivity between these conditions may appear to be odds with the finding that multivariate pattern similarity in EVC was significantly reduced for predictive vs. nonpredictive actions even immediately after training, and with previous MVPA findings in which classifier accuracy was at chance in both the hippocampus and EVC for nonpredictive actions while above chance for predictive actions³. Critically, however, background connectivity and MVPA are differentially sensitive to prediction in this task. Specifically, although participants cannot accurately predict outcomes of nonpredictive actions, they may nonetheless *inaccurately* predict outcomes. For example, the less predictable transitions for these cues may encourage hypothesis testing or other attempts to continue learning, or they may be predicting both outcomes associated with the cue (which each still co-occur 50% of the time, far higher than any other outcome). Less differentiated patterns in visual cortex may in fact reflect less differentiated neural predictions, as opposed to a lack of prediction. Likewise, in any of these cases, a multivariate classifier will seek evidence of the correct outcome, and so performance will be at chance on average. However, to the extent that background connectivity between the hippocampus and visual cortex reflects the process of prediction, whether accurate or inaccurate, it may be enhanced for both predictive and nonpredictive actions.

Verbal predictions for nonpredictive actions before and after each fMRI scan—and their relationship across participants to background connectivity—support the idea that the hippocampus may at first generate spurious predictions for nonpredictive events. While participants were required to be 100% accurate in identifying outcomes of predictive actions, there were no objectively correct or incorrect responses for nonpredictive actions. However, participants were more than 90% consistent on average in matching nonpredictive cues and actions to specific unpredictable outcomes immediately after training and were significantly less consistent in making such predictions for nonpredictive actions after the 3-day delay. Moreover, the small subset of participants who were still 100% consistent in their verbal predictions for nonpredictive actions after the 3-day delay exhibited significantly stronger background connectivity during nonpredictive events than did participants who made inconsistent predictions.”

Minor comments

Subiculum was not included in the hippocampal ROI. This is fine, as the authors’ hypotheses about which subfields should matter are motivated by their prior work. However, the subiculum is a large portion of the hippocampus and its exclusion should be noted somewhere earlier in the paper than the methods.

We thank the reviewer for pointing this out. In the caption for Figure 4A (which displays the hippocampal ROI of an example participant), we now describe and motivate how we defined the hippocampal ROI.

Results: Figure 4 caption, p. 13, line 2:

“The hippocampal seed and ROI included CA2/3, dentate gyrus, and CA1 (but not the subiculum) since these subfields were linked to pattern completion during action-based prediction in a prior study with the same task³.”

pg. 11: In reporting stats for the interaction for the cross correlation between EVC and lagged HPC time series, a t stat is reported that should likely be a F stat.

We thank the reviewer for pointing out this error. We have corrected the reporting of this statistic in the revised Results, on page 16.

Results: Time-lagged background connectivity, p. 16, line 7:

“In contrast, no such interaction was found when the hippocampus was lagged with respect to EVC ($F(1,22) = 0.04, p = .85$; Fig. 6B)”

It is unclear which part of each trial was included in the GLM parameters estimates for the multivariate patterns used in the representational similarity analysis. Were patterns based on models of the cue onset and duration or were the action and outcome also included in the model?

We thank the reviewer for prompting us to provide more detail on how the parameter-estimate patterns were calculated for the pattern similarity analyses (a suggestion also made by Reviewer #2). The GLM parameter estimates used in the similarity analyses were based on models of the complete cue-action-outcome sequences. Specifically, a double-gamma HRF was convolved with boxcar functions in which each trial was approximately 2.5 seconds in duration (depending on the mean response time for the participant and condition). The Methods section for multivariate pattern similarity (on page 34) now includes these and other details.

Methods: Multivariate pattern similarity, p. 34, line 15:

“Each cue-outcome transition was modeled with its own regressor and temporal derivative, constructed by convolving a boxcar function that matched the average trial duration for the condition (between 2188 and 2643 ms, depending on the participant’s mean response time) with a double-gamma hemodynamic response function.”

Reviewers' Comments:

Reviewer #1:

Remarks to the Author:

The authors have sufficiently addressed all of my previous concerns.

Reviewer #2:

Remarks to the Author:

The authors have thoughtfully responded to the concerns I raised in my initial review (as well as the concerns raised by other reviewers). The new analyses/data generally alleviate the concerns I had flagged. There is a greater reliance on between-subjects correlations than would be ideal (particularly since the sample size is modest), but because these are mostly supplementary analyses, I think this is ok. The revisions to the Introduction, Methods, and Discussion have also improved the manuscript. Overall, there are some interesting results that have stood up to scrutiny and should be of broad interest to readers.

Reviewer #3:

Remarks to the Author:

The revision by Hindy and colleagues offers a thorough response to my and the other reviewers' comments. The clarifications to the methods, as well as a comprehensive evaluation of the potential confounds noted during the review process satisfy my original concerns. I appreciate the authors' commitment to considering the reviewers' suggestions. The broader framing in the introduction and more in-depth consideration of the findings in the discussion appropriately place the contribution in the literature. As I mentioned in my original review, I think these findings will impact the field and the revisions only strengthen this belief.

Response to Reviewer #1

The authors have sufficiently addressed all of my previous concerns.

We thank the reviewer once again for their very helpful feedback and are pleased to have addressed their concerns.

Response to Reviewer #2

The authors have thoughtfully responded to the concerns I raised in my initial review (as well as the concerns raised by other reviewers). The new analyses/data generally alleviate the concerns I had flagged. There is a greater reliance on between-subjects correlations than would be ideal (particularly since the sample size is modest), but because these are mostly supplementary analyses, I think this is ok. The revisions to the Introduction, Methods, and Discussion have also improved the manuscript. Overall, there are some interesting results that have stood up to scrutiny and should be of broad interest to readers.

We are pleased that the new analyses addressed the reviewer's concerns and we agree that they significantly strengthened the manuscript. We are grateful to the reviewer for their useful feedback.

Response to Reviewer #3

The revision by Hindy and colleagues offers a thorough response to my and the other reviewers' comments. The clarifications to the methods, as well as a comprehensive evaluation of the potential confounds noted during the review process satisfy my original concerns. I appreciate the authors' commitment to considering the reviewers' suggestions. The broader framing in the introduction and more in-depth consideration of the findings in the discussion appropriately place the contribution in the literature. As I mentioned in my original review, I think these findings will impact the field and the revisions only strengthen this belief.

We thank the reviewer for their thoughtful ideas on how to broaden the framing of the paper and we agree that these revisions will significantly strengthen the impact of the work.